# Generalizable Hand-Object Modeling from Monocular RGB Images via 3D Gaussians

**Xingyu Liu**[*], **Pengfei Ren**[*], **Qi Qi, Haifeng Sun, Zirui Zhuang,**
**Jing Wang, Jianxin Liao, Jingyu Wang**[†]
State Key Laboratory of Networking and Switching Technology,
Beijing University of Posts and Telecommunications
{liuxingyu, rpf, qiqi8266, hfsun, zhuangzirui,
wangjing, liaojx, wangjingyu}@bupt.edu.cn

## Abstract

Recent advances in hand-object interaction modeling have employed implicit representations, such as Signed Distance Functions (SDF) and Neural Radiance Fields (NeRF) to reconstruct hands and objects with arbitrary topology and photo-realistic detail. However, these methods often rely on dense 3D surface annotations, or are tailored to short clips constrained in motion trajectories and scene contexts, limiting their generalization to diverse environments and movement patterns. In this work, we present **HOGS**, an adaptively perceptive 3D Gaussian Splatting (3DGS) framework for generalizable hand-object modeling from unconstrained monocular RGB images. By integrating photometric cues from the visual modality with the physically grounded structure of 3D Gaussians, HOGS disentangles inherent geometry from transient lighting and motion-induced appearance changes. This endows hand-object assets with the ability to generalize to unseen environments and dynamic motion patterns. Experiments on two challenging datasets demonstrate that HOGS outperforms state-of-the-art methods in monocular hand-object reconstruction and photo-realistic rendering.

## 1 Introduction

Fine-grained hand-object modeling is crucial for immersive AR/VR applications. Existing methods largely rely on dense 3D annotations or pre-scanned object models [16, 35, 47, 48, 15, 14], which incur high labeling costs and limit scalability. Leveraging the ubiquity and accessibility of monocular RGB images to reconstruct interactions, by contrast, offers a more practical avenue for seamless integration into consumer-grade AR/VR ecosystems. Recent works [50, 7] achieve hand-object reconstruction from short RGB clips but remain restricted to fixed environments and limited motion trajectories, requiring scene-specific optimization and retraining for new conditions. We present a cross-scene and cross-motion generalizable paradigm for photo-realistic hand-object modeling from monocular RGB images, reducing novel-scene setup from hours to seconds without post-training adaptation.

Recent studies have extensively explored implicit representations, such as signed distance functions (SDF) and Neural Radiance Fields (NeRF) [29], to advance fine-grained hand-object modeling. SDF-based methods [20, 3, 2, 27] densely optimize the zero-level set of a spatially continuous signed-distance field, to capture high-density hand-object surface with arbitrary topology. Neural implicit approaches [12, 31, 7] continuously encode scenes as volume density and view-dependent

---

[*] Equal contribution.
[†] Corresponding author.

39th Conference on Neural Information Processing Systems (NeurIPS 2025).

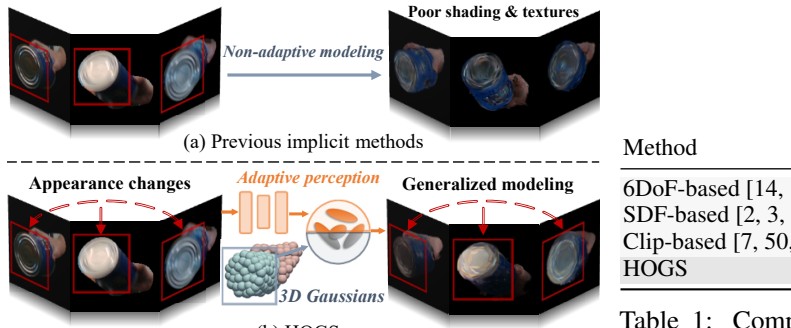

(a) Previous implicit methods

(b) HOGS

Figure 1: HOGS overcomes the transient appearance changes of hand-objects through the perceptual capabilities of neural networks, to achieve photo-realistic hand-object modeling with motion and scene adaptation.

| Method | Category-agnostic | 3D annotation-free | Cross-scene/motion |
|---|---|---|---|
| 6DoF-based [14, 15] | ✗ | ✗ | ✓ |
| SDF-based [2, 3, 35] | ✓ | ✗ | ✓ |
| Clip-based [7, 50, 37] | ✓ | ✓ | ✗ |
| HOGS | ✓ | ✓ | ✓ |

Table 1: Comparison of representative hand-object modeling methods across key desiderata. Our method is both category-agnostic and 3D surface annotation-free, while achieving strong generalization across environments and motions.

radiance, showing the feasibility of recovering photo-realistic hand-objects under temporally and spatially constrained conditions (e.g., from short video clips).

Despite these advances, existing SDF-based methods heavily rely on dense 3D surface supervision, while acquiring such precise annotations is typically impractical and cost-prohibitive in the wild. Additionally, previous neural implicit approaches suffer from two fundamental limitations. First, models trained on single short video sequences capture appearance and motion biases, as the visual conditions and motion patterns within such limited data are highly constrained; second, they neglect explicit perception of physical effects like hand-object contact and transient illumination phenomena. Persistent occlusions, dynamic shadows, and specular reflections inherent to real-world interactions violate the simplified assumptions of these methods, resulting in degraded geometry estimates and rendering artifacts, as illustrated in Fig. 1. As a result, they struggle to generalize beyond their constrained training clips when exposed to unseen environments or complex motion patterns.

To address these challenges, we explore how generalizable priors on appearance variation and motion dynamics can be learned from large-scale monocular RGB images, promoting hand-object representations that generalize across diverse environments and motion patterns. While implicit radiance fields achieve high photometric fidelity, they model scenes as undifferentiated volumes, lacking physically grounded structure for contact-level reasoning. Motivated by the need for both learnability and physical awareness, we adopt 3D Gaussian Splatting (3DGS) [21], where each Gaussian primitive encodes physically meaningful attributes. We integrate neural perception with this geometric representation to capture the underlying patterns of lighting and surface appearance, and to explicitly reason about hand-object contact and motion. This adaptively perceptive representation learning design endows hand-object assets with robust cross-scenario generalization capabilities.

To this end, we propose Hand-Object Gaussian Splatting (**HOGS**) for interacting hand-object reconstruction from monocular RGB images, with a central focus on generalization across diverse environments and motion sequences. Built upon an adaptively perceptive 3DGS-based representation, HOGS models articulated interactions via deformable Gaussian primitives driven by hand poses and object 6-DoF transformations. For generalizable hand-object modeling, we introduce two modules: a Vision-driven generalizable Perception Module (V-PM), which disentangles 3D Gaussians into geometry-invariant canonical templates and vision-dependent learnable components, explicitly decoupling appearance-invariant persistent geometry from transient photometric variations; and a Geometry-driven Pose refinement Module (G-PM), which employs a lightweight 3D neural network to extract geometric priors from 3D Gaussian primitives, utilizing the physical awareness to accurately refine hand-object pose and contact. HOGS jointly optimizes hand-object Gaussians and the proposed submodules from monocular images through a unified differentiable rendering pipeline, leveraging 2D photometric supervision without relying on dense 3D surface annotations.

Experimental results on two challenging datasets show that our method outperforms the state-of-the-art (SOTA) methods in monocular hand-object reconstruction and photo-realistic rendering. Furthermore, qualitative results and multimedia supplementary materials highlight its generalization

across diverse visual conditions and motion patterns. Code is available at `https://github.com/ru1ven/HOGS`. Our contributions could be summarized as:

- We propose a hand-object Gaussian splatting framework for interacting hand-object photo-realistic reconstruction from monocular RGB images, without relying on dense 3D surface annotations.

- We integrate visual and geometric priors from neural networks into the modeling of hand-object Gaussians, yielding hand-object assets with perceptual generalization across various environments and motions.

- Experiments demonstrate that our method significantly improves both reconstruction and rendering performance, showcasing strong generalization capabilities.

## 2  Related Work

### 2.1  Interacting Hand-object Reconstruction

Many works have been proposed to understand hand-object interactions [20, 2, 3, 50, 40, 48, 26, 32, 7, 16, 4, 9], with many focus on joint hand-object or hand-held object reconstruction. Early mainstream efforts [14, 15, 47, 40, 48] assume known object templates and use parametric models (e.g., MANO) with a fixed resolution, reducing mesh reconstruction to pose estimation. To achieve category-agnostic reconstruction, Hasson et al. [16] use AtlasNet [10] to deform object vertices from a sphere. Karunratanakul et al. [20] introduced SDF-based implicit fields for fine-grained hand-object reconstruction. AlignSDF [3] further integrates the strengths of parametric models and SDF by encoding pose priors into the implicit field. Chen et al. [2] use kinematic and temporal features to guide SDF-based 3D reconstruction. However, these methods rely on accurate and dense 3D annotations. With the development of geometric volume rendering techniques, several 2D photometric-supervised hand-object reconstruction methods have been proposed. MOHO [52] leverages occlusion-aware synthetic pre-training to pursue hand-held object reconstruction from a single-view image. Some recent methods [7, 37] attempt to exploit temporal coherence in monocular videos as a proxy for multi-view supervision. However, natural interaction sequences often exhibit persistent occlusions and invariant lighting patterns between the hand and object. These intrinsic limitations lead to a domain gap between observed motion patterns and unconstrained monocular scenarios. In consequence, prior methods struggle to bridge such a gap across scenarios with significant appearance-motion discrepancies, failing to robustly model interactions under varying motion and unseen scenarios.

### 2.2  Human-centric Photo-realistic Modeling

With the advancements in neural implicit and geometric representations, such as Neural Radiance Fields (NeRF) and 3D Gaussian Splatting (3DGS), significant achievements have been made in 3D scene reconstruction and novel view synthesis. Recently, some studies have explored adapting these representations to the photo-realistic rendering of dynamic human bodies and hands. Several works [45, 34, 5, 12, 31, 19, 36] leverage pose information derived from parametric models like SMPL [28] and MANO [38] to drive textured representations or neural fields for modeling dynamic humans or hands from multi-view inputs. HumanNeRF [44] further relaxes the requirements for multi-view inputs by adopting a simpler monocular setting, and proposes decoupling the motion field into skeletal rigid and non-rigid components. Subsequent studies [8, 18] focus on achieving significantly faster training speeds. More recently, some works have introduced 3DGS for animatable human modeling from monocular inputs. To capture fine details such as clothing and hair, HUGS [22] allows 3D Gaussians to deviate from the human body model. 3DGS-Avatar [37] proposes pose-dependent rigid and non-rigid deformations to handle highly articulated and out-of-distribution poses of clothed humans. GauHuman [17] refines human pose and Linear Blend Skinning (LBS) weights and employs human priors with a KL divergence measure for adaptive density control of 3D Gaussians. iHuman [33] binds 3D Gaussians to the human body surface for explicit normal rasterization, and optimization with normal supervision, to achieve 3D reconstruction and photo-realistic rendering. However, photo-realistic reconstruction of closely interacting and mutually occluding hands and objects from unconstrained monocular images remains challenging and underexplored.

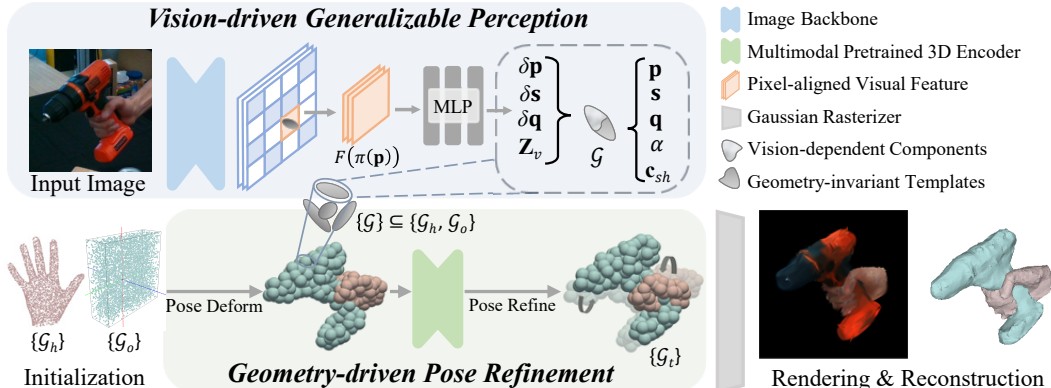

Figure 2: Our framework. We initialize deformable 3D Gaussians using hand articulations and object 6-DoF poses. The Vision-driven generalizable Perception Module (V-PM) disentangles Gaussians into geometry-invariant templates and vision-dependent components via extracted visual features, while the Geometry-driven Pose refinement Module (G-PM) extracts primitive-level 3D geometric features to refine hand-object poses and contacts. These modules are co-optimized in a differentiable rendering pipeline, jointly reconstructing photo-realistic geometry and appearance under cross-scene motion and illumination variations without per-sequence fine-tuning.

## 3 Method

Given a diverse collection of monocular RGB hand-object images across varied motions and scene context, HOGS jointly optimizes deformable 3D Gaussian primitives for each hand and each object, while learning generalizable interaction priors to refine hand-object modeling. During inference on unseen sequences involving novel interactions and appearances, HOGS can adaptively modulate the trained hand-object assets in a purely feed-forward manner, without extra fine-tuning.

Fig. 2 illustrates the overview of our framework. We use pose-deformable 3D Gaussians to model hand-object interaction from monocular images across multiple sequences (Sec. 3.1). To enable generalizable hand-object Gaussian modeling, we introduce a Vision-driven generalizable Perception Module (V-PM), to dynamically adapt 3D Gaussians to visual scene variations (Sec. 3.2), as well as a Geometry-driven Pose refinement Module (G-PM), extracting Gaussian primitive-level geometric features to enhance robustness against inaccurate hand-object motion estimation (Sec. 3.3). Details of optimization and mesh extraction are provided in the Appendix.

### 3.1 Hand-Object Interaction Modeling

We represent a hand and object in the canonical coordinate space through a set of 3D Gaussian primitives $\{\mathcal{G}_h\}$ and $\{\mathcal{G}_o\}$, and their positions are initialized by random sampling on the canonical MANO surface and within the 3D bounding box, respectively. Each 3D Gaussian $\mathcal{G}$ is defined by its center $\mathbf{p} \in \mathbb{R}^3$, covariance matrix $\mathbf{\Sigma} \in \mathbb{R}^{3\times3}$, opacity $\alpha$ and view-dependent color $\mathbf{c}$. Its geometry is parameterized as

$$\mathcal{G}(\mathbf{x}) = \exp\left(-\frac{1}{2}(\mathbf{x} - \mathbf{p})^T \mathbf{\Sigma}^{-1}(\mathbf{x} - \mathbf{p})\right),\tag{1}$$

where the covariance matrix $\Sigma = \mathbf{R}\mathbf{S}\mathbf{S}^T\mathbf{R}^T$ is represented by a scaling matrix $\mathbf{S} \in \mathbb{R}^{3\times3}$ and a rotation matrix $\mathbf{R} \in \mathbb{R}^{3\times3}$, which are represented by the diagonal vector $\mathbf{s} \in \mathbb{R}^3$ and a quaternion vector $\mathbf{q} \in \mathbb{R}^4$ in practice. During the rendering process, 3D Gaussians are projected onto the camera view. Given a viewing transformation $W$ and the Jacobian matrix of the projective transformation $J$, the covariance matrix $\Sigma'$ in camera coordinates is computed as: $\Sigma' = JW\Sigma W^T J^T$. To account for the mutual occlusion relationships between the hand and the object during rasterization, their Gaussians are jointly accumulated via alpha blending to compute the pixel color $C$:

$$C = \sum_{i \in \mathcal{N}_{ho}} \mathbf{c}_i \alpha_i \prod_{j=1}^{i-1}(1 - \alpha_j),\tag{2}$$

where $\mathcal{N}_{\text{ho}}$ is the depth-ordered set of hand and object Gaussians overlapping the pixel.

To adapt to the motion of the hand and object, we use a pose-driven approach to transform the canonical Gaussian $\mathcal{G}$ to the pose-transformed Gaussian $\mathcal{G}_t$ in the observation space. For the hand branch, similar to prior methods [37], we apply the forward Linear Blend Skinning (LBS) to transform the center position $\mathbf{p}$ and rotation matrix $\mathbf{R}$ of each 3D Gaussian:

$$\mathbf{p}_t = \left( \sum_{i=1}^{n_b} w_i(\mathbf{p}) \cdot B_i \right) \cdot \mathbf{p}, \tag{3}$$

$$\mathbf{R}_t = \left( \sum_{i=1}^{n_b} w_i(\mathbf{p}) \cdot B_i \right)_{1:3,1:3} \cdot \mathbf{R}, \tag{4}$$

where $\mathbf{p}_t$ and $\mathbf{R}_t$ are the center and rotation matrix of the transformed 3D Gaussian, $\{B_i\}_{i=1,\ldots,n_b}$ are the bone transformations derived from the hand poses $\theta$, and $\{w_i(\mathbf{p})\}_{i=1,\ldots,n_b}$ are the learnable skinning weights at position $\mathbf{p}$. Similarly, for the object branch, we apply a 6D rigid transformation based on the object rotation and translation to transform the canonical Gaussian to the posed Gaussian. In practice, we use an off-the-shelf regressor [35] to initialize the hand pose and object 6-DoF pose.

## 3.2  Vision-driven Generalizable Perception

Across multiple motion sequences, variations in the visual environment result in changes in the appearance of the hand and object, such as alterations in lighting intensity and direction, transient lighting patterns induced by surface materials and occlusions, as well as motion-induced blur and texture deformation. To enable the 3D Gaussian model to generalize across various visual environments in unconstrained monocular scenarios, we decouple the parameters of the vanilla Gaussian primitives into geometry-invariant templates and vision-dependent components.

Take color $\mathbf{c}$ as an example, the vanilla 3DGS uses Spherical Harmonics (SH) coefficients for view-relevant appearance modeling, which is insufficient for capturing the aforementioned visual variations across scenes. To address this, we first employ a Vision Transformer [6] as the image backbone to extract visual features $F$ from the hand-object region, and obtain pixel-aligned visual features $F(\pi(\mathbf{p}_t))$ using bilinear sampling based on the 2D projection location $\pi(\mathbf{p}_t)$ of each pose-transformed 3D Gaussian position $\mathbf{p}_t$. We use an MLP to encode the visual features and Gaussian parameters into a vision-dependent appearance component $\mathbf{Z}_v$:

$$\mathbf{Z}_v = \psi(F, F(\pi(\mathbf{p}_t)), \mathcal{G}), \tag{5}$$

where the 3D coordinates of each Gaussian within $\mathcal{G}$ are represented using a multi-resolution hash grid encoding [30]. Then, we compute the vision-dependent color component in an adaptive manner. Specifically, we use an MLP network $\psi_1$ to obtain a color weight vector $\alpha$ from the vision-dependent appearance component $\mathbf{Z}_v$ and the precomputed color $\mathbf{c}_{sh}$ derived from the SH coefficients. The weight $\alpha$ quantifies the importance of different visual features in determining the final appearance, which can be formulated as:

$$\alpha = Sigmoid((\psi_1(\mathbf{Z}_v, \mathbf{c}_{sh})), \tag{6}$$

The final color $\mathbf{c}$ of each 3D Gaussian is obtained by weighting the SH color and the visual component:

$$\mathbf{c} = (1 - \alpha) \cdot \mathbf{c}_{sh} + \alpha \cdot Sigmoid(\mathbf{Z}_v). \tag{7}$$

For the remaining 3D Gaussian parameters, we follow a similar process, but in a simpler manner:

$$(\delta\mathbf{p}, \delta\mathbf{s}, \delta\mathbf{q}) = \psi_2(F, F(\pi(\mathbf{p}_t)), \mathcal{G}), \tag{8}$$

$$\mathbf{p}' = \mathbf{p} + \delta\mathbf{p}, \tag{9}$$

$$\mathbf{s}' = \mathbf{s} \cdot \exp(\delta\mathbf{s}), \tag{10}$$

$$\mathbf{q}' = \mathbf{q} \cdot [1, \delta q_1, \delta q_2, \delta q_3], \tag{11}$$

where $\delta\mathbf{p}$, $\delta\mathbf{s}$, and $\delta\mathbf{q}$ are the vision-dependent components of Gaussian center, scale, and rotation, respectively, and $\mathbf{p}'$, $\mathbf{s}'$, and $\mathbf{q}'$ are the corresponding updated Gaussian parameters.

## 3.3 Geometry-driven Pose Refinement

Sequence-specific reconstruction methods independently apply structure-from-motion (SfM) to each video for initial motion estimation, while per-frame cues are used to refine poses. This per-sequence design tightly couples the pipeline to the specific trajectories and scene layouts of each sequence. As a result, the learned motion and contact patterns are sequence-specific and do not generalize well to unseen interactions or motions beyond the observed trajectories.

To address monocular images with previously unseen motion patterns, we leverage the rich geometric and physical properties of 3D Gaussians, which encode physically meaningful spatial and volumetric information and provide explicit geometric cues for guiding motion and interaction reasoning. Consequently, we employ a lightweight 3D neural network to integrate this information, achieving precise hand-object pose refinement and contact reasoning.

Firstly, to capture the geometric structure information of 3D Gaussians, a straightforward approach is to treat the Gaussian primitives as a 3D point cloud and extract geometric features using point cloud networks. However, it may exhibit lower efficiency when handling non-geometric attributes like color. To address this limitation, we adopt ULIP [46], a multi-modal pre-trained model, to improve the comprehensive understanding of 3D Gaussians by effectively integrating both geometric and visual information. Specifically, we employ PointNet++ [14], pre-trained by ULIP, as the 3D backbone to encode the pose-transformed Gaussian centers $\mathbf{p}_t$ and Gaussian parameters $\mathcal{G}$ into the subsampled Gaussian centers $\mathbf{p}_t' \in \mathbb{R}^{N' \times 3}$ and a 3D Gaussian feature matrix $\mathbf{F}_{\mathcal{G}} \in \mathbb{R}^{N' \times C}$. This process enables efficient feature extraction and downsampling while maintaining the structural and physical properties of the 3D Gaussian representation. Then, following [14], we employ a Transformer encoder containing self-attention modules [41] and a 3-layer MLP to obtain the hand-relative object translation offset $\Delta T_o \in \mathbb{R}^3$ and object rotation offset $\Delta R_o \in \mathbb{R}^{3 \times 3}$.

Additionally, we employ a contact and penetration loss [16] to optimize interactions between the hand and object. We treat the centers of hand and object Gaussians as hand-object point clouds, and following [16], apply distance-based penetration and collision losses between them to enforce physically plausible contact and prevent interpenetration. Leveraging these geometric attributes of hand-object 3D Gaussians, our approach relaxes the need for ground-truth object templates or detailed 3D vertex annotations required by existing contact optimization methods [16, 49, 20], while optionally utilizing 6D pose annotations. Details of training losses are provided in the Appendix.

# 4 Experiment

## 4.1 Datasets

**DexYCB** [1] is a hand-object dataset containing 582K RGB-D frames over 1,000 sequences of 10 subjects grasping 20 different objects. We follow the dataset split in [35], filtering samples without interactions, obtaining 147,526 training samples. For reconstruction evaluation, we follow [2, 52, 27] to downsample the video data to 6 frames per second, resulting in 5,928 testing samples.

**HO3D_v3** [13] is an RGB-D hand-object interaction dataset with 10 subjects manipulating 10 objects from the YCB dataset. Following the evaluation protocol of [7], we select 18 sequences for training and evaluate the quality of reconstructed hand-held object meshes.

## 4.2 Baselines

We conduct comparisons with existing hand-object and hand-held object reconstruction methods, including 3D dense supervised baselines (typically SDF-based) and 2D photometric supervised baselines (e.g., MOHO [52] and HOLD [7]). Additionally, we re-implement 3DGS-Avatar [37] and GOF [51], which originally utilize 3DGS to model animable human bodies and reconstruct static unbounded scenes, respectively, extending them to a hand-object baseline and a rigid object baseline (marked by †). In particular, we adapt 3DGS-Avatar by replacing SMPL with MANO for the hand branch and substituting pose-dependent deformations (e.g., LBS) with rigid transformations for the object branch.

| Method | $CD_h\downarrow$ | $F_h@1\uparrow$ | $F_h@5\uparrow$ | $CD_o\downarrow$ | $F_o@5\uparrow$ | $F_o@10\uparrow$ |
|---|---|---|---|---|---|---|
| *3D Supervised Methods:* | | | | | | |
| Hasson et al. [16] | 0.537 | 0.115 | 0.647 | 1.94 | 0.383 | 0.642 |
| Grasping Field [20] | 0.364 | 0.154 | 0.764 | 2.06 | 0.392 | 0.660 |
| AlignSDF [3] | 0.358 | 0.162 | 0.767 | 1.83 | 0.410 | 0.679 |
| gSDF [2] | **0.302** | **0.177** | **0.801** | 1.55 | 0.437 | 0.709 |
| HORT [4] | - | - | - | - | 0.630 | 0.850 |
| *2D Supervised Methods:* | | | | | | |
| MOHO [52] | - | - | - | - | 0.600 | 0.810 |
| GOF$^\dagger$ [51] | - | - | - | 0.68 | 0.610 | 0.834 |
| Ours | 0.481 | 0.133 | 0.732 | **0.24** | **0.785** | **0.918** |

Table 2: Quantitative results of hand-object reconstruction on DexYCB. Video data is downsampled to 6 frames per second.

| Method | $CD_o\downarrow$ | $F_o@10\uparrow$ | $CD_{pose}\downarrow$ |
|---|---|---|---|
| iHOI [49] | 3.8 | 75.8 | 41.7 |
| DiffHOI [50] | 4.3 | 68.8 | 43.8 |
| HOLD [7] | **0.4** | **96.5** | 11.3 |
| Ours | 0.7 | 96.2 | **2.7** |

Table 3: Quantitative results of hand-held object reconstruction on HO3D.

| Method | PSNR↑ | SSIM↑ | LPIPS↓ |
|---|---|---|---|
| GOF$^\dagger$ [51] | 29.58 | 0.9686 | 31.69 |
| 3DGS-Avatar$^\dagger$ [37] | 29.71 | 0.9690 | 30.52 |
| Ours | **31.12** | **0.9728** | **26.83** |

Table 4: Quantitative results of hand-object photo-realistic rendering on DexYCB.

### 4.3 Implementation Details

We initialize the hand Gaussians and object Gaussians by randomly sampling $K = 5,000$ points within the canonical MANO surface and the 3D bounding box, respectively. For optimization, we follow [21] to employ cloning or splitting and pruning to adaptively control the density of the 3D Gaussians during optimization. We employ an individual set of Gaussians for each subject or object. For the input of the visual encoder, we crop the hand-object region of the RGB image and resize it to 224×224. We use an AdamW optimizer [24] for training. On DexYCB, we train the model for a total of 360k iterations, which takes approximately 10 hours on an NVIDIA RTX 4090 GPU. On HO3D, we train for 200k iterations. After 360k iterations on DexYCB, we fix the parameters of the 3D Gaussians and continue training for an additional 10 epochs, focusing solely on optimizing the pose parameters to prevent underfitting.

### 4.4 Metrics

**Geometric Metrics.** For geometric evaluation of the object, we follow [7] to report the Chamfer distance ($\mathbf{CD}_o$) in $cm^2$ after ICP alignment with rotation, translation, and scaling, Chamfer distance incorporating 6-DoF pose ($\mathbf{CD}_{pose}$), as well as the F-score evaluated at thresholds of 5mm and 10mm ($\mathbf{F}_o@\mathbf{5}$ and $\mathbf{F}_o@\mathbf{10}$). For hand reconstruction, we report the Chamfer distance ($\mathbf{CD}_h$) and F-score at 1mm and 5mm thresholds ($\mathbf{F}_h@\mathbf{1}$ and $\mathbf{F}_h@\mathbf{5}$). In addition, to evaluate hand-held object pose performance, we report Object Center Error (**OCE**), Mean Corner Error (**MCE**), and the standard pose estimation metric average closest point distance (**ADD-S**) following [35].

**Rendering Metrics.** For evaluation of hand-object photo-realistic rendering, we report the peak signal-to-noise ratio (**PSNR**), structural similarity index (**SSIM**) [43], and learned perceptual image patch similarity (**LPIPS**) [53]. Only the region within the hand-object mask is considered. Note that LPIPS values in all tables are scaled up by 1000.

### 4.5 Comparisons with State-of-the-arts

**Surface Reconstruction.** To validate the quality of hand-object modeling, we compare with hand-object and hand-held object reconstruction SOTAs on the DexYCB dataset. As shown in Table 2, our method significantly improves object reconstruction quality over SOTA methods while achieving

| Method | OCE↓ | MCE↓ | ADD-S↓ |
|---|---|---|---|
| AlignSDF [3] | 27.0 | - | - |
| Wang et al. [42] | 27.3 | 32.6 | 15.9 |
| Lin et al. [25] | 39.8 | 45.7 | 31.9 |
| gSDF [2] | 19.1 | - | - |
| HOISDF [35] | 18.4 | 27.4 | 13.3 |
| Ours | **18.1** | **26.1** | **12.0** |

Table 5: Hand-held object pose estimation results on DexYCB.

| Method | CJ | PSNR↑ | SSIM↑ | LPIPS↓ |
|---|---|---|---|---|
| 3DGS-Avatar [37] | ✗ | 29.71 | 0.9690 | 30.52 |
| | ✓ | $29.48_{\downarrow0.23}$ | $0.9689_{\downarrow0.0001}$ | $32.02_{\uparrow1.50}$ |
| Ours (w/o V-PM) | ✗ | 29.57 | 0.9687 | 31.75 |
| | ✓ | $29.32_{\downarrow0.25}$ | $0.9685_{\downarrow0.0002}$ | $33.44_{\uparrow1.69}$ |
| Ours | ✗ | 31.12 | 0.9728 | 26.83 |
| | ✓ | $31.00_{\downarrow0.12}$ | $0.9727_{\downarrow0.0001}$ | $27.57_{\uparrow0.74}$ |

Table 6: Ablation study for the impact of color jittering on DexYCB. *CJ* represents Color jitter.

| ID | Method | PSNR↑ | SSIM↑ | LPIPS↓ | OCE↓ | MCE↓ | ADD-S↓ |
|---|---|---|---|---|---|---|---|
| 0 | Full model | 31.12 | **0.9728** | **26.83** | **17.83** | **25.95** | **11.87** |
| 1 | w/o contact optimization | 31.09 | 0.9727 | 26.93 | 18.10 | 26.13 | 11.97 |
| 2 | w/o pre-trained 3D backbone | 31.09 | 0.9726 | 26.96 | 18.11 | 26.40 | 11.88 |
| 3 | w/o G-PM | **31.17** | 0.9721 | 27.39 | 18.45 | 27.37 | 13.31 |
| 4 | w/o adaptive appearance | 29.92 | 0.9700 | 29.95 | - | - | - |
| 5 | w/o V-PM | 29.57 | 0.9687 | 31.75 | - | - | - |

Table 7: Ablation study on DexYCB. Both rendering quality and pose results are shown.

comparable hand reconstruction performance to previous 3D dense supervised methods. The hand-held reconstruction performance comparison with SOTA methods on HO3D is shown in Table 3. Our method achieves pose-independent reconstruction quality ($\mathbf{CD}_o$ and $\mathbf{F}_o@\mathbf{10}$) comparable to SOTA methods without requiring per-sequence optimization, while significantly improving pose-dependent reconstruction accuracy ($\mathbf{CD}_{pose}$), showcasing our enhanced generalization capability across diverse hand-object motion. Furthermore, we conduct hand-held object pose estimation comparisons in Table 5, and our method achieves superior object pose accuracy, validating the effectiveness of the proposed geometry-driven pose refinement strategy. Fig. 4 is the qualitative results. Our method produces more plausible object shapes compared to gSDF [2], avoiding collapses and deformations. In contrast to GOF [51], our method can better recover local geometric structures and mitigate mesh holes and incompleteness, demonstrating superior Gaussian modeling quality.

**Photo-realistic Rendering.** We perform a comparative analysis of photo-realistic rendering of hand objects on DexYCB. As shown in Table 4, our method outperforms baselines on all the metrics by a large margin. Qualitative comparisons are shown in Fig. 3. Compared to 3DGS-Avatar [37], our method achieves superior color and texture fidelity and enhanced hand-object pose accuracy. Notably, ours demonstrates robust capabilities in both modeling hand-object shadow patterns (column 2) and adapting to chromatic variations caused by material reflectance under dynamic illumination conditions (columns 3-4).

## 4.6 Ablation Study

**Impact of visual environment changes.** To further validate the generalization capability of HOGS against visual environment variations, we simulate appearance alterations induced by diverse illumination patterns on the DexYCB dataset and compare the rendering quality impacts between HOGS and baseline methods. Specifically, we apply a random color jittering within the range of [-0.3, 0.3] to each RGB image, which is used for photometric supervision and rendering evaluation. We conduct comparative analyses on rendering quality under color jittering conditions across three configurations: our full model, our model without V-PM, and 3DGS-Avatar. As demonstrated in Table 6, while color jittering minimally affects structural distortion in both 3DGS-Avatar and ours without V-PM, it significantly degrades PSNR and LPIPS metrics due to pixel color errors and image perceptual discrepancies. In contrast, our approach incorporates visual generalization awareness to substantially mitigate rendering quality degradation, demonstrating effective adaptation to visual environment

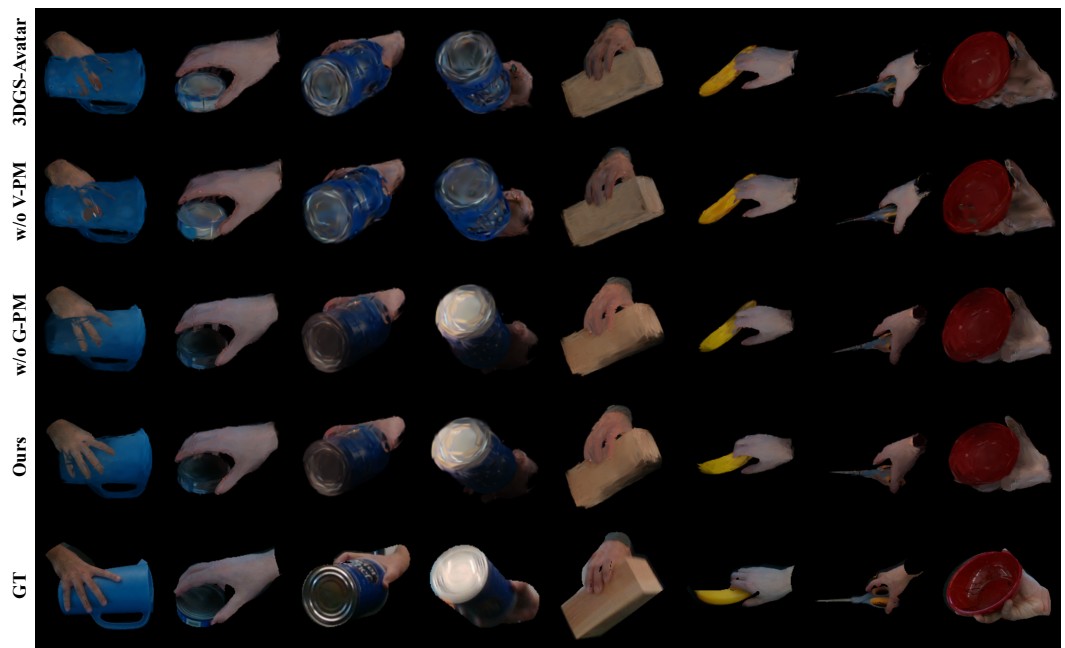

Figure 3: Qualitative results of hand-object photo-realistic rendering on DexYCB.

| ID | $F$ | $F(\pi(\mathbf{p}_t))$ | $\mathcal{G}$ | PSNR↑ | SSIM↑ | LPIPS↓ |
|----|-----|------------------------|---------------|-------|-------|--------|
| 0 |  |  |  | 29.57 | 0.9687 | 31.75 |
| 1 | ✓ |  | ✓ | 30.49 | 0.9703 | 29.66 |
| 2 | ✓ | ✓ |  | 30.52 | 0.9704 | 29.50 |
| 3 |  | ✓ | ✓ | 31.08 | 0.9725 | 27.31 |
| 4 | ✓ | ✓ | ✓ | **31.12** | **0.9728** | **26.83** |

Table 8: Ablation of the vision-driven components on DexYCB. $F$ represents global visual features, $F(\pi(\mathbf{p}_t)$ represents pixel-aligned visual features, and $\mathcal{G}$ represents 3DGS parameters.

changes. Furthermore, Fig. 5 intuitively shows that our method can adaptively capture diverse global color tones for rendering, exhibiting remarkable robustness to visual appearance variations.

**Ablation on generalizable modules.** We perform ablation studies to verify the effectiveness of two proposed critical components, V-PM and G-PM. As shown in Table 7, our full model demonstrates superior performance compared to configurations excluding hand-object contact optimization (ID 1) or multi-modal pre-trained 3D backbone integration (ID 2). The proposed G-PM yields significant enhancements in pose results (ID 0 vs. ID 3). Notably, although G-PM does not directly optimize 3D Gaussians, the hand-object pose refinement improves rendering quality, particularly SSIM-measured structural fidelity and LPIPS-based perceptual quality. Additionally, the removal of either the adaptive appearance modeling component (ID 4) or the V-PM (ID 5) results in significant degradation of rendering quality. The qualitative ablation is shown in Fig. 3. The V-PM can capture hand-object lighting and shadow patterns (columns 2-5), while the G-PM effectively alleviates hand-object penetration artifacts (column 1) and refines imprecise object pose (columns 6-7).

**Ablation on the vision-driven components.** To assess the impact of different features in the vision-driven component of 3DGS, we conduct an ablation study on DexYCB using global visual features $F$, pixel-aligned visual features $F(\pi(\mathbf{p}_t))$, and 3DGS parameters $\mathcal{G}$. As shown in Table 8, discarding the vision-driven component altogether (ID 0) results in the lowest rendering quality. Incorporating any single feature improves performance, and the improvement is particularly notable when the fine-grained pixel-aligned visual features are included. The best results are achieved when all three features are used (ID 4), indicating that global context, pixel-level alignment, and 3DGS parameters provide complementary information that jointly enhances reconstruction fidelity.

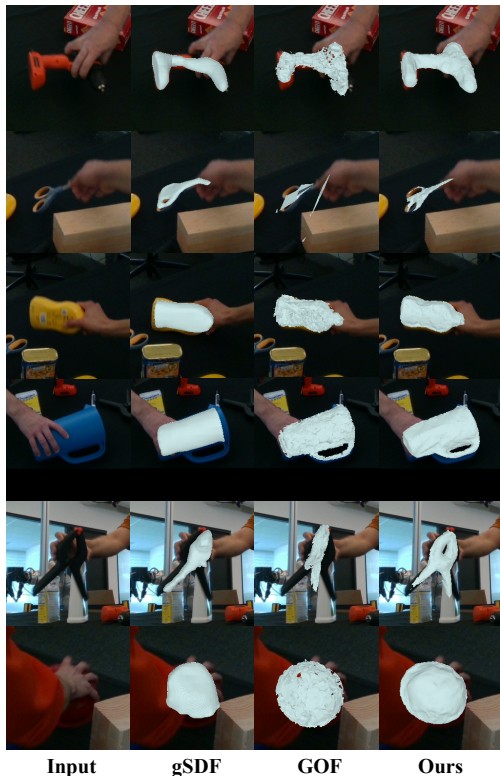

| Input | gSDF | GOF | Ours |

Figure 4: Qualitative results of hand-held object reconstruction on DexYCB.

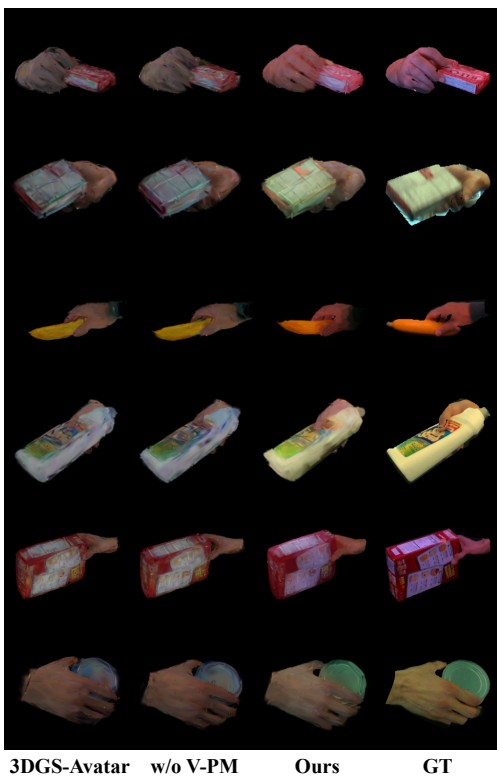

| 3DGS-Avatar | w/o V-PM | Ours | GT |

Figure 5: Qualitative ablation for the impact of color jittering on hand-object rendering quality.

# 5   Conclusion

In this paper, we propose a 3DGS-based interacting hand-object modeling framework from unconstrained monocular RGB images. The key insight lies in leveraging neural networks to endow 3D Gaussians with appearance and motion adaptability, guided by generalizable visual and geometric perceptual cues. Extensive experiments demonstrate the effectiveness of HOGS in achieving fine-grained interacting hand-object modeling, significantly advancing monocular hand-object reconstruction and photo-realistic rendering performance.

**Limitation.** While HOGS exhibits strong cross-scene and cross-motion generalization, its performance is constrained by the diversity of object categories encountered during training. Consequently, it focuses on hand-object assets that generalize to unseen visual contexts and motion patterns, rather than synthesizing entirely new object categories. A promising direction for future work is large-scale training on extensive hand-object datasets, enabling zero-shot generalization to novel object categories.

# Acknowledgments

This work was supported in part by the National Natural Science Foundation of China under Grants (62406039, 62471055, U23B2001, 62321001, 62101064, 62171057, 62071067), the High-Quality Development Project of the MIIT(2440STCZB2584), the Ministry of Education and China Mobile Joint Fund (MCM20200202, MCM20180101), BUPT Excellent Ph.D. Students Foundation (CX20241014), the Project funded by China Postdoctoral Science Foundation (2023TQ0039, 2024M750257, GZC20230320), the Fundamental Research Funds for the Central Universities (2024PTB-004), the 2025 Education and Teaching Reform Project Funding at Beijing University of Posts and Telecommunications (2025YZ005).

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

## Appendix

In Appendix contents, we provide:

- Details of mesh extraction (Section A).
- Optimization of hand-object Gaussians (Section B).
- Discussion of mesh alignment (Section C).
- More qualitative results (Section D).

Video supplementary materials are provided as separate files.

## A  Mesh Extraction

Inspired by the impressive novel view synthesis capabilities of 3D Gaussian Splatting , recent studies [11, 51] have focused on exploring the use of 3D Gaussian representations for surface reconstruction. To extract explicit meshes from hand-object Gaussians, we extend Gaussian Opacity Fields (GOF), a 3DGS-based surface reconstruction method tailored for unbounded scenes, into an interacting hand-object mesh reconstruction framework.

### A.1  Preliminary: Gaussian Opacity Fields

GOF uses an explicit ray-Gaussian intersection instead of projection, which allows evaluating the opacity value or transmittance of any 3D point $\mathbf{x}$ along the ray. At the most basic level, given a single 3D Gaussian $\mathcal{G}_k$, the opacity at any point along a ray can be defined as:

$$O_k(\mathcal{G}_k, \mathbf{o}, \mathbf{r}, t) = \begin{cases} \mathcal{G}_k^{1D}(t) & \text{if } t \leq t^* \\ \mathcal{G}_k^{1D}(t^*) & \text{if } t > t^* \end{cases} \tag{12}$$

where $\mathbf{x} = \mathbf{o} + t\mathbf{r}$. Intuitively, the opacity increases until it reaches its maximal value, and remains constant thereafter. Next, considering a set of 3D Gaussians $\mathcal{G}$, the opacity of point $\mathbf{x}$ is given by:

$$O(\mathbf{o}, \mathbf{r}, t) = \sum_{k=1}^{K} \alpha_k O_k(\mathcal{G}_k, \mathbf{o}, \mathbf{r}, t) \prod_{j=1}^{k-1} (1 - \alpha_j O_j(\mathcal{G}_j, \mathbf{o}, \mathbf{r}, t)) \tag{13}$$

As a 3D point might be visible by any training view, the vanilla GOF defines the opacity field $O(x)$ of a 3D point $x$ as the minimal opacity value among all training views or viewing directions:

$$O(x) = \min_{(\mathbf{o}, \mathbf{r})} O(\mathbf{o}, \mathbf{r}, t) \tag{14}$$

### A.2  Dynamic Hand-Object Reconstruction

However, for dynamic hand-object reconstruction, directly employing GOF faces two problems: (1) We observe the interacting hand-object from a single viewing direction, rather than scanning various views in an unbounded scene, making it impossible to rely on multiple input view directions to model the opacity field; (2) Unlike static unbounded scenes, the hand-object motions cause instability in the position and rotation of the 3D Gaussians, resulting in drastic changes in the local geometry. To address these issues, we first generate inward-oriented rays that converge toward the geometric core of the hand-object, enabling systematic surface interrogation. Specifically, we uniformly sample $K = 16$ points $\{\mathbf{o}_k\}_{k=1}^{K}$ from the surface of a bounding sphere closely encapsulating the hand-object. For each sampled point $\mathbf{o}_k$, a directed ray $\mathbf{r}_k$ is parametrized as:

$$\mathbf{r}_k(t) = \mathbf{o}_k + t \cdot \frac{\mathbf{c} - \mathbf{o}_k}{\|\mathbf{c} - \mathbf{o}_k\|}, \quad t \geq 0 \tag{15}$$

Next, we adopt a smoother manner to integrate the opacity of each 3D point from different views. Specifically, we enforce hard opacity constraints while maintaining smoothness in visible regions. The conditional branch serves two critical purposes: (1) **Physical Constraint.** Zero opacity is strictly enforced when any viewing ray confirms full transparency, preventing phantom geometry artifacts;

(2) **View Consensus.** The mean operation in non-transparent regions smooths out inconsistencies from sparse view sampling. The final opacity field can be defined as:

$$O(\mathbf{x}) = \begin{cases} 0 & \text{if } \exists\, (\mathbf{o_k}, \mathbf{r_k}) \in \Omega, \\ & \quad O(\mathbf{o_k}, \mathbf{r_k}, t) = 0 \\ \frac{1}{|\Omega|} \sum_{(\mathbf{o_k}, \mathbf{r_k}) \in \Omega} O(\mathbf{o_k}, \mathbf{r_k}, t) & \text{otherwise} \end{cases} \tag{16}$$

Finally, following [51], we use the center and corners of 3D bounding boxes around the 3D Gaussian primitives as vertex sets for the tetrahedral mesh, and utilize the Marching Tetrahedra algorithm [39] for triangle mesh extraction upon assessing the opacity at tetrahedral points.

# B    Optimization

## B.1    Pixel Color Loss

Beyond joint rendering for interaction modeling, we adopt independent color supervision for hands and objects, to address the color bleeding issue in close interaction regions where traditional unified rendering fails to disentangle component-specific appearances:

$$C = \sum_{i \in \mathcal{N}_{\text{ho}}} \mathbf{c}_i \alpha_i \prod_{j=1}^{i-1} (1 - \alpha_j)$$

$$C_{\text{hand}} = \sum_{i \in \mathcal{N}_{\text{ho}}} \mathbf{c}_i^h \alpha_i \prod_{j=1}^{i-1} (1 - \alpha_j), \quad \mathbf{c}_i^h = \begin{cases} \mathbf{c}_i & i \in \mathcal{N}_{\text{hand}} \\ 0 & \text{otherwise} \end{cases} \tag{17}$$

$$C_{\text{obj}} = \sum_{i \in \mathcal{N}_{\text{ho}}} \mathbf{c}_i^o \alpha_i \prod_{j=1}^{i-1} (1 - \alpha_j), \quad \mathbf{c}_i^o = \begin{cases} \mathbf{c}_i & i \in \mathcal{N}_{\text{obj}} \\ 0 & \text{otherwise} \end{cases}$$

where $\mathcal{N}_{\text{ho}} = \text{DepthSort}(\mathcal{N}_{\text{hand}} \cup \mathcal{N}_{\text{obj}})$ ensures correct occlusion handling across components. The pixel-level RGB loss can be formulated as:

$$\mathcal{L}_{\text{rgb}} = \|C - C^{\text{gt}}\|_1 + \|C_{\text{hand}} - C_{\text{hand}}^{\text{gt}}\|_1 + \|C_{\text{obj}} - C_{\text{obj}}^{\text{gt}}\|_1 \tag{18}$$

## B.2    Mask Loss

To facilitate geometry supervision, we compute the opacity value by accumulating the alpha values, performed separately for hand, object, and jointly hand-object:

$$O = \sum_{i \in \mathcal{N}_{\text{ho}}} \alpha_i \prod_{j=1}^{i-1} (1 - \alpha_j)$$

$$O_{\text{hand}} = \sum_{i \in \mathcal{N}_{\text{ho}}} \alpha_i^h \prod_{j=1}^{i-1} (1 - \alpha_j), \quad \alpha_i^h = \begin{cases} \alpha_i & i \in \mathcal{N}_{\text{hand}} \\ 0 & \text{otherwise} \end{cases} \tag{19}$$

$$O_{\text{obj}} = \sum_{i \in \mathcal{N}_{\text{ho}}} \alpha_i^o \prod_{j=1}^{i-1} (1 - \alpha_j), \quad \alpha_i^o = \begin{cases} \alpha_i & i \in \mathcal{N}_{\text{obj}} \\ 0 & \text{otherwise} \end{cases}$$

The mask loss measures the L1 distance between the rendered opacity and the corresponding ground truth mask values:

$$\mathcal{L}_{\text{mask}} = \|O - M\|_1 + \|O_{\text{hand}} - M_{\text{hand}}\|_1 + \|O_{\text{obj}} - M_{\text{obj}}\|_1 \tag{20}$$

## B.3    Perceptual Loss

To enhance high-frequency detail preservation and mitigate blurring artifacts in synthesized inter-actions, we extend the independent supervision paradigm to perceptual feature space, optimizing

| Method | $\text{CD}_o\downarrow$ | $\text{F}_o@5\uparrow$ | $\text{F}_o@10\uparrow$ |
|---|---|---|---|
| Hasson et al. [16] | 1.94 | 0.383 | 0.642 |
| Grasping Field [20] | 2.06 | 0.392 | 0.660 |
| AlignSDF [3] | 1.83 | 0.410 | 0.679 |
| gSDF [2] | 1.55 | 0.437 | 0.709 |
| Ours (not rotation-aligned) | 1.51 | 0.497 | 0.730 |
| Ours | **0.24** | **0.785** | **0.918** |

Table 9: Comparison of object reconstruction on DexYCB.

LPIPS as the perceptual loss with AlexNet [23] as the backbone. Following the RGB loss structure, we employ LPIPS metric for decomposed components. The final perceptual loss aggregates multi-component measurements:

$$
\begin{aligned}
\mathcal{L}_{\text{perc}} = \ &\text{LPIPS}\left(\mathcal{R}(C), \mathcal{R}(C^{\text{gt}})\right) \\
&+ \text{LPIPS}\left(\mathcal{R}(C_{\text{hand}}), \mathcal{R}(C_{\text{hand}}^{\text{gt}})\right) \\
&+ \text{LPIPS}\left(\mathcal{R}(C_{\text{obj}}), \mathcal{R}(C_{\text{obj}}^{\text{gt}})\right)
\end{aligned}
\tag{21}
$$

where $\mathcal{R}(\cdot)$ denotes the rendering function from alpha-composited colors to RGB images.

### B.4 Pose Loss

To ensure physically plausible hand-object interactions, we supervise the 6D pose parameters (rotation $\mathbf{R} \in \text{SO}(3)$, translation $\mathbf{t} \in \mathbb{R}^3$) and the corner positions $\mathbf{P}_C$ of manipulated objects through the SmoothL1 loss:

$$
\begin{aligned}
\mathcal{L}_{\text{rot}} &= \text{SmoothL1}\left(\hat{\mathbf{R}}, \mathbf{R}^{\text{gt}}\right) \\
\mathcal{L}_{\text{trans}} &= \text{SmoothL1}\left(\hat{\mathbf{t}}, \mathbf{t}^{\text{gt}}\right) \\
\mathcal{L}_{\text{corner}} &= \text{SmoothL1}\left(\hat{\mathbf{P}}_C, \mathbf{P}_C^{\text{gt}}\right)
\end{aligned}
\tag{22}
$$

The overall pose loss $\mathcal{L}_{\text{pose}}$ is then computed as the weighted sum of these components:

$$
\mathcal{L}_{\text{pose}} = \lambda_{\text{rot}}\mathcal{L}_{\text{rot}} + \lambda_{\text{trans}}\mathcal{L}_{\text{trans}} + \lambda_{\text{corner}}\mathcal{L}_{\text{corner}}
\tag{23}
$$

where the weight factors $\lambda_{\text{rot}}$, $\lambda_{\text{trans}}$, and $\lambda_{\text{corner}}$ are set as 10, 1e4, and 1e3, respectively. In our framework, the ground truth object poses are optional. For example, in comparison on the HO3D dataset, since HOLD [7] does not use any pose annotations, our method does not access the ground truth poses in this experiment either.

### B.5 Overall Loss

In addition to the aforementioned loss functions, we further introduce several regularization terms to enhance the robustness and physical plausibility of our framework. Specifically, we follow [37] to incorporate a Skinning Loss $\mathcal{L}_{\text{skin}}$ for regularizing the forward skinning network. To preserve local geometric consistency during deformation, we employ an as-isometric-as-possible constraint, which consists of $\mathcal{L}_{\text{iso-pos}}$ and $\mathcal{L}_{\text{iso-cov}}$ to restrict neighboring 3D Gaussian centers and covariance matrices, ensuring they maintain similar distances after deformation. Furthermore, we treat the centers of hand and object Gaussians as hand-object point clouds, and following [16], apply distance-based penetration loss $\mathcal{L}_{\text{pen}}$ and contact loss $\mathcal{L}_{\text{cont}}$ between them to enforce physically plausible contact and prevent interpenetration. The overall loss can be formulated as:

$$
\begin{aligned}
\mathcal{L}_{\text{total}} = \ &\lambda_{\text{rgb}}\mathcal{L}_{\text{rgb}} + \lambda_{\text{mask}}\mathcal{L}_{\text{mask}} + \lambda_{\text{perc}}\mathcal{L}_{\text{perc}} + \lambda_{\text{pose}}\mathcal{L}_{\text{pose}} \\
&+ \lambda_{\text{cont}}\mathcal{L}_{\text{cont}} + \lambda_{\text{pen}}\mathcal{L}_{\text{pen}} + \lambda_{\text{skin}}\mathcal{L}_{\text{skin}} \\
&+ \lambda_{\text{iso-pos}}\mathcal{L}_{\text{iso-pos}} + \lambda_{\text{iso-cov}}\mathcal{L}_{\text{iso-cov}},
\end{aligned}
\tag{24}
$$

where $\lambda_{\text{rgb}}$, $\lambda_{\text{mask}}$, $\lambda_{\text{perc}}$, $\lambda_{\text{pose}}$, $\lambda_{\text{cont}}$, $\lambda_{\text{pen}}$, $\lambda_{\text{skin}}$, $\lambda_{\text{iso-pos}}$, and $\lambda_{\text{iso-cov}}$ are set to 1, 0.1, 0.01, 1, 20, 10, 0.1, 1, and 100, respectively.

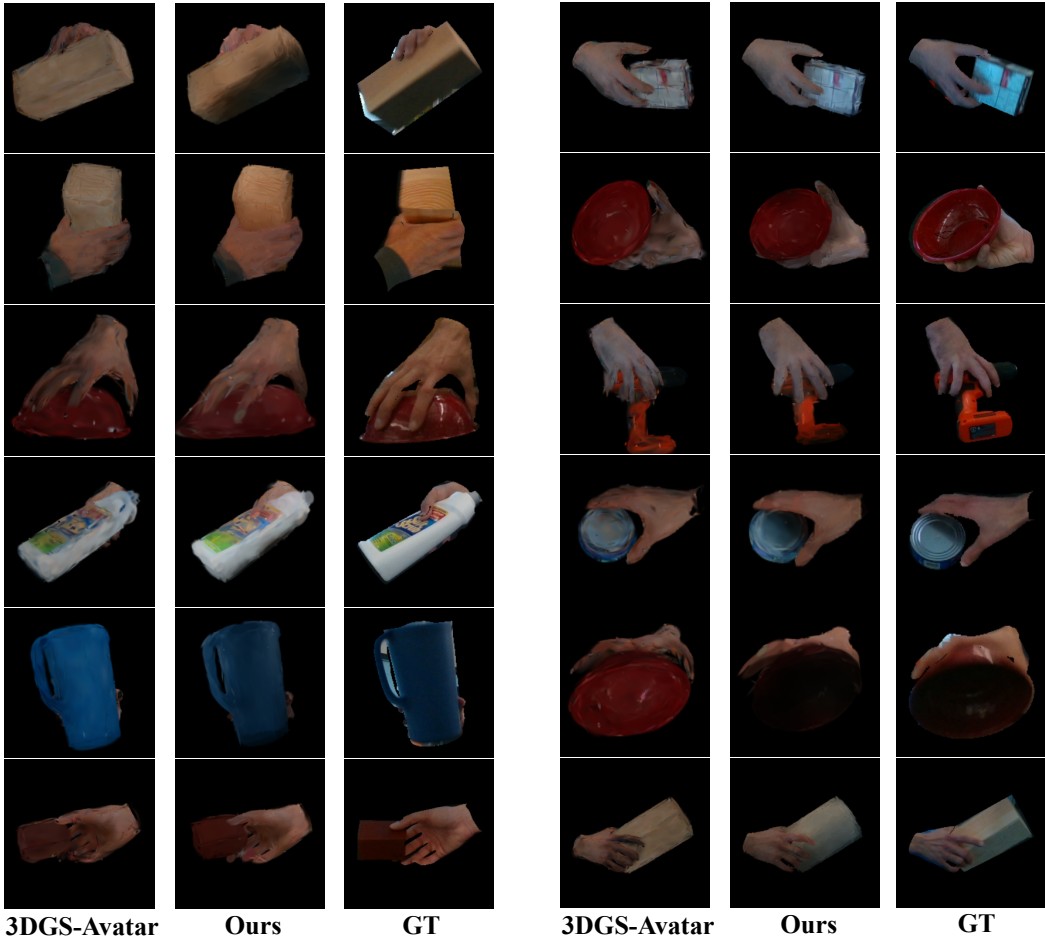

| 3DGS-Avatar | Ours | GT | 3DGS-Avatar | Ours | GT |

Figure 6: Qualitative results on DexYCB dataset.

## C  Discussion of Mesh Alignment

Existing SDF-based methods [3, 20, 2] reconstruct object meshes within a unit bounding box using non-uniform scaling and evaluate pose-independent reconstruction via ICP with translation and scaling. In contrast, 6-DoF pose-driven methods [7] use ICP with translation, scaling, and rotation for alignment. Table 9 shows that our method outperforms SDF-based approaches under both alignment strategies.

## D  Qualitative Results

We present more qualitative comparisons of hand-object rendering on DexYCB in Fig. 6. The rendering results of our method show more delicate colors and more robust hand-object poses, and can adapt to the light and shadow patterns between hand and object.

## E  Societal Impacts

This paper proposes a fine-grained hand–object interaction modeling framework from monocular RGB images, with the positive impact of broadening access to immersive technologies in AR/VR, HCI, and robotics applications. A possible negative impact may arise from privacy and fairness concerns due to implicit data collection and dataset bias, necessitating cautious and transparent deployment in real-world applications.

