# OpenReview forum: "Generalizable Hand-Object Modeling from Monocular RGB Images via 3D Gaussians"
_NeurIPS.cc/2025/Conference — NeurIPS 2025 poster_

### Official Review · Reviewer_bphE · 2025-06-28

**Clarity:** 1
**Significance:** 2
**Originality:** 3
**Rating:** 4
**Confidence:** 4

**Summary:**

This paper presents HOGS, a novel pipeline aims to tackle photorealistic hand-object rendering that are generalizable to unseen motions and environments. The key components of the system are two modules: (i) a vision-driven perception that augments the learning of 3D-GS parameters with per-pixel features encoded by a vision transformer, and (ii) a geometry-driven pose refinement module that further improves hand-object poses via pretrained point cloud encoders. These two designs lead to superior appearance and geometry accuracy over both implicit field based methods and pose estimation methods.

**Questions:**

1. The ablation study seems to only demonstrate quantitative improvement, can authors show more qualitative improvement regarding (i) the effects of weight quantity over vision dependent component Z for Eq.(7), (ii) the refined poses after geometry-driven pose refinement

2. How is the vision transformer in L165 used? is it pretrained or also learned from the two benchmarks?

3. The authors are also encouraged to clarify on the set-up of input, i.e. how is the cross-scene actually defined for the input? e.g. number of scenes involved and how are they partitioned.

4. Can the authors provide some more clarification on inference time analysis, in particular the time for training the GS model and how is the adaption to unseen scenes with 0.1s achieved in L27?

**Ethical Concerns:**

["NO or VERY MINOR ethics concerns only"]

**Final Justification:**

See below discussions.

**Limitations:**

yes

**Quality:**

3

**Strengths And Weaknesses:**

1. The major concern of the work is its generalizability presented in the current paper, in particular:

(i) Note that the method rely on template-based object pose estimator to obtain the 6DoF object poses, which limits its generalizablity to a few objects only (e.g. those in DexYCB and HO3D) and perhaps are not feasible to test on in-the-wild videos and objects.

(ii) The set-up of the current model is also not clear, in particular how is "cross-scene, cross-motion" in L25 achieved? Does the model specifically trained with a mixed of environments only or trained with mixed objects as well?

(iii) The environment change shown in qualitative result is actually limited to different sequences of the same benchmark only. Since HO3D and DexYCB share the same object at completely difference, how is it possible to generalize across benchmarks?

2. The supplementary video also shows limited visual quality, in L40 - L42 the authors mentioned some challenging issues such as dynamic shadow (interaction-dependent) and specular effects (interaction-independent), that must be resolved with an in-depth modeling of hand-object contact. However, I do not see these points being tackled in the video other than fidelity improvement.

Overall I think the current paper is a bit over-claimed, e.g. the work clearly can not handle arbitrary unconstrained RGB images as the tile indicates, and its generalizability is questionable. However, I do find the quality is superior than the baseline methods, so I am willing to increase the score if the authors are able to present more generalization evaluation.

---

> ### Author Rebuttal · Authors · 2025-07-29
>
> We sincerely appreciate the reviewer’s insightful comments. To facilitate a more accurate interpretation of our work’s scope and contributions, we would like to provide some clarifications that may help address potential misunderstandings.
>
> **Clarification of Our Setting**
> HOGS focuses on ***hand-object assets generalizing to unseen scenes and motions rather than synthesizing entirely new object categories***. In particular, prior methods, such as HOLD, are typically tailored to fixed motion trajectories and environments, spanning only short clips of a few hundred frames and requiring per‑sequence modeling. In contrast, HOGS generalizes hand‑object assets across novel motions and lighting conditions (e.g., diverse interactions, poses, and dynamic shadows) without any per‑sequence fine‑tuning. During inference, we run on monocular images from unseen sequences with novel viewpoints, environments, and motions, rather than on new object categories. Details follow.
>
> - **Object Pose Estimating Generalization**
> While template-based monocular pose estimators may limit category generalization, this is not a concern in our setting, which focuses on cross-sequence generalization for trained hand and object assets. On the contrary, compared to prior methods that adopt per-sequence motion estimation (e.g., SfM), our monocular pose estimators significantly improve generalization by eliminating the need for sequence-specific optimization.
>
> - **Cross-scene and Cross-motion Generalization**
> HOGS is trained end-to-end on a large collection of monocular interaction sequences that include mixed objects, motions, and environments. While the training data indeed encompasses multiple objects, it focuses primarily on variations in interaction behaviors and environmental context. Specifically, on DexYCB, we sample 147,526 monocular images from over 1,000 sequences covering 10 subjects and 20 objects. Through such training, HOGS acquires scene modulation capability and, when encountering unseen sequences with novel contacts, poses, and lighting conditions, can adaptively modulate hand-object assets from monocular images in a purely feed-forward manner without per-sequence optimization.
>
> - **Clarification on Evaluation Scope and Object Categories**
> Our goal is not novel-object synthesis but cross-sequence generalization for trained hand and object assets. At inference time, we evaluate on unseen sequences featuring novel motions and viewpoints. Since our method is not intended for novel-category synthesis, we focus on benchmarks that share YCB object categories rather than those containing entirely new objects.
>
> **Transient Appearance Effects Visualization**
> As demonstrated in Figure 3 of the main paper, our visually-driven modulation effectively captures fine-grained transient phenomena such as hand-induced shadows on the can surface (columns 2–4) and dynamic reflections, which static baseline methods fail to reproduce. These effects are also observable in the supplementary video, where the can’s movement induces dynamic, transient surface reflections. Due to the transient nature of these changes, they may be overlooked; we will include more explicit annotations to better showcase them.
>
> **Qualitative Effects of V-PM and G-PM**
> As shown in Figures 3 (rows 2–4) and 5, we provide qualitative ablations of V-PM and G-PM: removing the vision-dependent component leads to degraded rendering quality and reduced robustness to color jitter, while removing G-PM affects geometric consistency. However, we only visualized the effect of G-PM on the rendered results. We appreciate the reviewer’s suggestion and will include explicit 6D pose visualizations (e.g., via 3D bounding boxes) in the final version to more clearly illustrate the benefits of geometry-driven pose refinement.
>
> **Details of ViT**
> We use a CLIP-pretrained ViT backbone and fine-tune it with an initial learning rate of 1e-6 during HOGS training.
>
> **Input Setup and Cross-Scene Definition**
> The input setup and cross-scene definition are detailed above in the ‘Cross-Scene and Cross-Motion Generalization’ subsection within ‘Clarification of Our Setting’. We appreciate the reviewer’s feedback and will incorporate clarifications in the final version to prevent any possible confusion.
>
> **Inference Efficiency**
> Training on DexYCB takes approximately 20 hours, as described in Sec. 4.3. '0.1s adaptation' means HOGS adapts to unseen scenes purely via feed-forward inference without fine-tuning, achieving **11.6 FPS rendering** on NVIDIA 4090 GPU. Explicit mesh reconstruction requires additional post-processing, taking seconds. By contrast, HOLD requires hours of per‑sequence optimization and lacks real‑time rendering.
>
> **Clarification on Wording and Claims**
> We regret any confusion caused by the term 'unconstrained'. Our intention was to emphasize that, unlike prior sequence-tailored methods requiring per-sequence fine-tuning, our approach generalizes to unseen monocular images in a purely feed-forward manner. We will revise the wording and clarify the setting in the final version to avoid overstating the claims.

---

> > ### Comment · Reviewer_bphE · 2025-08-04
> > **Rebuttal Reply**
> >
> > Thank you for your detailed reply and clarification. I aggree the paper can benefit from incorporating clarification of above details and wordings.
> >
> > However, The motivation remains unclear to me. Since the authors justified that "generalization" means seen objects with unseen views or configurations, how is this set up actually differs from the majority of works in hand object pose estimation, *e.g.* [32, 28] as compared by the authors? These works are also jointly trained on benchmarks across multiple YCB objects and can feedforwardly adapt to unseen secnarios without test-time fine-tuning. What is the paper's technical novelty compared to them, other than improvements on performance?

---

> > > ### Author Response · Authors · 2025-08-04
> > > **Response to Generalization Concern**
> > >
> > > We appreciate your insightful question regarding *“generalization”*. We'd like to clarify both the motivation and technical positioning of our work in relation to prior hand-object pose estimation methods and recent NeRF-based pipelines. ﻿
> > >
> > >
> > > First, prior methods you mentioned [32, 38] rely heavily on pre-defined object templates [12, 37, 38] or densely annotated surfaces and vertex supervision [2, 3, 14, 18, 32]. However, acquiring such 3D ground-truth annotations is computationally expensive, which limits their scalability and practicality. In contrast, our method offers two key advantages:
> > > (i) We achieve higher-fidelity modeling without relying on dense 3D annotations or object templates;
> > > (ii) Rather than estimating only 6D poses or coarse meshes, we reconstruct high-quality hand-object assets with realistic textures, shading, and lighting effects, enabling photorealistic rendering and richer visual reasoning.
> > > ﻿
> > >
> > > In addition, to clarify, the *“generalization”* we emphasize is not directed at these traditional methods, but rather at recent NeRF/3DGS-based pipelines such as HOLD. While HOLD shows the potential of using photometric supervision for high-quality asset construction, it is tailored to specific sequences (e.g., ~300 frames per clip) that cannot directly generalize to novel motions and interaction contexts. Novel sequence requires hours of scene-specific optimization, making the method impractical for scalable or real-time use. ﻿
> > > By contrast, our method introduces a **feedforward Gaussian modeling pipeline** with **scene-conditioned modulation**, enabling 3D-annotation-free and fine-tuning-free real-time inference. Despite this efficiency, our method achieves quality competitive with per-sequence optimization approaches like HOLD.

---

> > > > ### Comment · Reviewer_bphE · 2025-08-04
> > > > **Rebuttal Reply**
> > > >
> > > > Thank you for further replies, most of my concerns are addressed. After further reviewing this paper, I am willing to increase the score to borderline accept.
> > > >
> > > > Having said that, I still encourage the authors to (i) perform a major revision on the phrasing about contributions, method setting, detailed abaltion, and discussion to related works, and (ii) discuss limitation in terms of generalizability to unseen objects, which will be helpful for highlighting the novelty of the work.

---

> > > > > ### Author Response · Authors · 2025-08-05
> > > > >
> > > > > Thank you for your constructive feedback and for reconsidering your score.
> > > > > In the final version, we will revise the phrasing of contributions and method setting, strengthen the ablation and related work discussion, and explicitly discuss the limitation on generalizing to unseen objects. We appreciate your valuable suggestions to help us improve the clarity and completeness of our work.

---

### Official Review · Reviewer_fKDv · 2025-07-01

**Clarity:** 3
**Significance:** 3
**Originality:** 3
**Rating:** 5
**Confidence:** 5

**Summary:**

The paper tackles hand-object reconstruction from a single monocular RGB video. It leverages deformable Gaussian epsilloids as the 3D representation. In order to capture the color change over time, it proposes to compute a vision-dependent color component from the pixel-aligned image features and the SH coefficients. Moreover, it leverages a lightweight 3D network to adjust the hand-relative pose offset. Experiments on DexYCB and HO3D demonstrates its effectiveness.

**Questions:**

Please refer to weaknesses. My main concern is about the statement of generalization ability, which requires addtional experiments to support. The failure to address this might lead to the concern of overclaiming. Despite this concern, I think the method is effective and thus recommend borderline accept as my initial rating.

**Ethical Concerns:**

["NO or VERY MINOR ethics concerns only"]

**Final Justification:**

My concern in generalization ability is addressed by the authors, so I raise my rating.

**Limitations:**

yes

**Quality:**

3

**Strengths And Weaknesses:**

Strengths:
1. It achieves strong performance in object reconstruction, pose estimation, photorealistic rendering on DexYCB.
2. Using deformable Gaussians for hand-object reconstruction is technically sound. The method to capture color change and refine object pose is simple and effective.

Weaknesses:
1. Generalization: The paper states that it learns generalizable priors that can be adapted to unseen environements and complex motion. However, the method is trained and tested on the same dataset (DexYCB, HO3D), which cannot support the generalization ability. The scene number and object number in both datasets is somewhat limited. Some objects in the test split are included in the training split.
2. The symbols and equations require careful revision. Why does the sigmoid function appear in Eq 7? Moreover, alpha in Eq 6 is usually considered as the opacity in Gaussian Splatting, please use another symbol instead.
3. No inference speed comparison is provided. Inference speed is an important factor for AR/VR applications.

---

> ### Author Rebuttal · Authors · 2025-07-29
>
> We appreciate the reviewer’s recognition of our method’s strong performance and technical soundness. We would like to clarify some misunderstandings regarding the intended scope and generalization of our work.
>
> **Clarification of Generalization Capability**
> HOGS focuses on generalizing hand-object assets to unseen scenes and motions rather than synthesizing entirely new object categories. Specifically, our goal is to improve over prior methods that are limited to fixed motion trajectories and environments (e.g., only hundreds of frames), by enabling assets to generalize across novel motions and lighting conditions. Thus, during inference, we evaluate on monocular images from novel sequences with unseen viewpoints, environments, and motions, while not on new object categories.
>
> **Symbol and Equation Clarity**
> We appreciate the reviewer’s suggestion to clarify the symbols, and will replace *σ* with *sigmoid*(·)  to avoid confusion with the opacity symbol. The sigmoid function in Eq. (7) is used as an activation for the color component, following the design used for other attributes in 3D Gaussians.
>
>
> **Inference Efficiency**
> HOGS adapts to unseen scenes purely via feed-forward inference without any fine-tuning, achieving a **`rendering speed of 11.6 FPS`**  on NVIDIA 4090 GPU. Explicit mesh reconstruction requires additional post-processing, which takes a few seconds. By contrast, HOLD requires hours of per‑sequence optimization and lacks real‑time rendering.

---

> > ### Comment · Reviewer_fKDv · 2025-08-05
> >
> > Thanks for your response. However, I still have concerns in the genealization ability. The experimental setting is the same as the previous methods, e.g., training and testing data is the same. Experiments presented in the paper does not support the claim of generalization, since the previous methods can generalize as well.

---

> > > ### Author Response · Authors · 2025-08-06
> > >
> > > We sincerely appreciate your valuable feedback. We respectfully clarify that our claim on generalization refers to the ability to ***scale assets to novel sequences without any per-sequence tuning***.
> > > ﻿
> > >
> > > Although Table 3 presents comparable reconstruction results, our method operates under a scalable setting, while others do not. Specifically, HOLD requires hours of optimization for each test sequence, as its motion estimation and pose refinement are conditioned on sequence-specific timestamp embeddings tied to training data. This restricts HOLD to seen trajectories and prevents direct inference on novel frames. In contrast, our method requires no test-time optimization and supports efficient feedforward inference at ~0.086s per frame. This explains why HOLD is only evaluated on small-scale clips (e.g., 18 sequences in HO3D) rather than datasets spanning a large number of sequences like DexYCB. This highlights a key gap in scalability and deployability between sequence-specific methods and ours.
> > > ﻿
> > >
> > > Additionally, in Tables 2 and 4, we compare with monocular reconstruction/rendering baselines ***under the same generalization setting*** (i.e., tested on novel frames without test-time tuning). Our method outperforms them without requiring dense 3D supervision.
> > > ﻿
> > >
> > > Lastly, we appreciate your comment regarding the limitation in generalizing to untrained object categories and will make an explicit discussion in the revised manuscript.

---

> > > > ### Comment · Reviewer_fKDv · 2025-08-06
> > > >
> > > > Thanks for your response. Now most of my concerns are addressed. I have no further questions.

---

### Official Review · Reviewer_9ss3 · 2025-07-01

**Clarity:** 1
**Significance:** 2
**Originality:** 2
**Rating:** 3
**Confidence:** 3

**Summary:**

This paper proposes HOGS, a 3D Gaussian Splatting (3DGS)-based framework for generalizable hand-object reconstruction from unconstrained monocular RGB images. While there are recent implicit function-based methods proposed for hand-object reconstruction, they rely on 3D surface annotations or limited to short clip inputs constrained in motions and scenes, with requiring scene-specific optimization that limits generalizability. To address this limitation, it integrates a Vision-driven generalizable Perception Module (V-PM), and Geometry-driven Pose refinement Module (G-PM) that are directly trained to infer visual and geometry-driven Gaussian Splat parameters, without per-sequence fine-tuning to achieve better generalizability. Experiments show that HOGS achieves competitive performance in monocular hand-object reconstruction and photorealistic rendering.

**Questions:**

See the Weaknesses section above.

**Ethical Concerns:**

["NO or VERY MINOR ethics concerns only"]

**Final Justification:**

During the author–reviewer discussion, some of my questions were addressed; however, my concerns regarding two of my initially raised weaknesses—**empirical performance of the proposed method** and **writing quality**—remain.

Regarding the **empirical performance of the proposed method**, I am concerned that it does not outperform existing baselines proposed for hand–object reconstruction (upper subtable of Table 2 and Table 3). The justifications provided during the author–reviewer discussion for this suboptimal performance were not very clear and did not align with the initial narrative presented in the main paper.

Related to these unclear arguments, my most serious concern is the **writing quality**. I find some of the technical arguments presented both in the paper and the rebuttal misleading—for example, repeatedly referring to 3DGS as an implicit representation in the main paper, or discussing *efficiency* and *appearance modeling* aspects in response to my question regarding suboptimal *shape reconstruction accuracy* compared to HOLD.

Therefore, I am leaning towards borderline rejection at this time, to ensure that the paper does not contain misleading technical arguments. Since the proposed method does have strengths (e.g., efficiency), I believe it could be a good fit for a future conference if the technical arguments are refined to be more reasonable and consistent.

**Limitations:**

Yes

**Quality:**

2

**Strengths And Weaknesses:**

[Strengths]

1. Good paper structure

The paper is well structured, and the teaser and method figures (Figures 1–2) are clearly presented.

2. Good research direction

I agree that the overall research direction—to develop a generalizable hand-object reconstruction method without requiring per-sequence training—is important for achieving higher generalizability and improved inference speed. However, I have some concerns regarding the current performance of the method.

[Weaknesses]

1. Empirical performance of the proposed method

In the main comparison results (Tables 2–3), the proposed method appears to perform suboptimally compared to the baselines. I am concerned that the statement "HOGS outperforms state-of-the-art methods in monocular hand-object reconstruction and photo-realistic rendering" (Lines 14–15) may be overstated. Additionally, the qualitative results—particularly the rendering results in Figures 3 and 5—appear very suboptimal, and in the video results, the proposed method looks worse than HOLD. While the method demonstrates better generalizability (e.g., smaller performance drop in color-modulated scenes, Table 6), I believe the main performance should still be comparable to existing state-of-the-art methods.

2. Choice of baselines

The latest 3D supervised method baseline considered in Table 2 is gSDF (CVPR 2023). Is there a reason more recent baselines (e.g., G-HOP [R1]) were not included in the comparison?

[R1] Ye et al., G-HOP: Generative Hand-Object Prior for Interaction Reconstruction and Grasp Synthesis, CVPR 2024.

3. Writing quality

The overall writing quality should be improved to meet the acceptance standard. The paper is generally not easy to read, and several statements either (1) misrepresent existing literature or (2) contain grammatical errors. Examples include:

  - (Lines 1–4, 101–102) The paper repeatedly describes 3DGS as an implicit representation, but it is an explicit representation.

  - (Lines 28–34) The introduction separates the characteristics of SDF-based methods from implicit approaches, even though SDF is a type of implicit representation.

  - (Line 220) "Implement Details" should be "Implementation Details".

  - (Line 106, minor comment) When referring to SMPL and MANO, citations are needed.

---

> ### Author Rebuttal · Authors · 2025-07-29
>
> We thank the reviewer for recognizing the paper’s clear structure and the importance of our research direction. Below, we address the concerns about the current performance.
>
> **Clarification of Empirical Performance**
> We appreciate the reviewer’s insightful feedback on both quantitative and qualitative results. **It is important to highlight that our method operates under a substantially different and more challenging setting.** Unlike methods such as HOLD that rely on per-sequence optimization tailored to specific scenes, our model is purely feed-forward during inference, requiring no optimization. This presents a more demanding scenario where the model must generalize across diverse motions and lighting conditions without any scene-specific adaptation. Despite this, we achieve comparable or better generalization than sequence-tailored methods (e.g., Table 3), and significantly outperform other baselines under the same setting (e.g., MOHO), which we believe underscores the effectiveness of our approach.
>
> **Choice of Baselines**
> Our comparisons focus primarily on recent 2D-supervised methods, such as MOHO (CVPR 2024) and 3DGS-Avatar (CVPR 2024). These methods leverage only 2D annotations (e.g., 2D keypoints, photometric supervision) and share a similar evaluation setting with ours. **Additionally, we compare against the latest 3D-supervised SOTA method, HORT (concurrent work, ICCV 2025 accepted)** [1], and our method, without requiring dense 3D ground-truth supervision, still demonstrates superior object reconstruction performance on DexYCB.
>
> | Methods | F@5↑ | F@10↑ |
> | :---    | :--- | :---  |
> | gSDF    | 0.44 | 0.71  |
> | HORT    | 0.63 | 0.85  |
> | Ours    | **0.79** | **0.92** |
>
> Regarding G-HOP (CVPR 2024), which pre-trains diffusion-based implicit hand-object priors for reconstructing interaction sequences, designed for video-based interaction reconstruction and cannot be directly used as a comparable baseline.
>
> **Writing Quality**
> We appreciate the reviewer’s suggestions and will improve the paper’s clarity and accuracy by:
> 1. Revising descriptions of implicit representations (e.g., NeRF and SDF) versus explicit representations (3DGS) for correctness.
> 2. Fixing typographical errors and adding missing citations.
>
> **Reference**
> [1] Chen Z. et al., "HORT: Monocular Hand-held Objects Reconstruction with Transformers," *ICCV*, 2025.

---

> > ### Comment · Reviewer_9ss3 · 2025-08-05
> >
> > I appreciate the authors' response and for sharing additional results. However, I have some remaining questions:
> >
> > **(1) Clarification of Empirical Performance**
> >
> > Although HOLD relies on optimization during monocular video-based reconstruction (to encourage hand-object contact while enforcing the alignment of 2D reprojections of the hand and object), I do not find this methodology to specifically hinder generalizability or to be tailored to particular scenes. In fact, as shown in Table 3, the proposed method does not demonstrate noticeably better performance or generalizability compared to HOLD—outperforming it in one metric but underperforming in two—which makes this argument unclear. It would be helpful if the authors could further clarify this point.
> >
> > **(2) Choice of Baselines**
> >
> > Thank you for sharing the additional results. Could the authors clarify why only a subset of evaluation metrics was reported—for example, why Chamfer Distance was omitted?

---

> > > ### Author Response · Authors · 2025-08-05
> > >
> > > **(1) Clarification of Empirical Performance**
> > >
> > >
> > > We respectfully clarify that our claim on *generalization* is not based solely on numerical performance, but rather on ***the scalability of directly running on unseen sequences without any per-sequence tuning***.
> > > ﻿
> > >
> > > In Table 3, HOLD requires hours of optimization independently for each test sequence (\~300 frames per clip), as its motion estimation and pose refinement ***are conditioned on sequence-specific timestamp embeddings***. These embeddings are tied to known training frames, preventing direct inference on unseen frames without retraining. In contrast, our method performs efficient, scalable feedforward inference at \~0.086 seconds per frame, without any fine-tuning.
> > >
> > >
> > > This also explains why HOLD is only evaluated on small-scale clips (e.g., 18 sequences on HO3D), but is not applicable to larger datasets like DexYCB: it cannot handle unseen frames across hundreds of sequences, and such extensive per-sequence optimization is impractical. This highlights a key gap in scalability and deployability between HOLD and our method.
> > > ﻿
> > >
> > > The comparison in Table 3 aims to demonstrate that our method, under a more challenging and scalable setting that does not rely on per-sequence optimization, achieves performance comparable to sequence-specific methods like HOLD.
> > > Meanwhile, Tables 2 and 4 highlight our method’s generalization ability compared with monocular reconstruction/rendering baselines on unseen images.
> > >
> > > **(2) Choice of Baselines**
> > >
> > > Because HORT reports the **mean** Chamfer Distance, while we report the **median**, a direct comparison may be misleading. For a comprehensive evaluation, we provide complete results below and will update the final version:
> > >
> > >
> > > | Methods |  CD$_{mean}$↓| CD$_{median}$↓| F@5↑ | F@10↑ |
> > > |---------|-------|-------|-------|-------|
> > > | GraspingField    | 4.5 | 2.06|0.39  | 0.66  |
> > > | AlignSDF    | 3.9 | 1.83|0.41  | 0.68  |
> > > | gSDF    | 3.4 | 1.55|0.44  | 0.71  |
> > > | HORT    |1.1  |- |0.63  | 0.85  |
> > > | Ours    |**0.9** | **0.24**| **0.79**  | **0.92**  |
> > >
> > > >Mean Chamfer Distance is more sensitive to outliers. Our method achieves superior reconstruction results across multiple metrics without relying on dense 3D supervision.

---

> ### Comment · Reviewer_9ss3 · 2025-08-06
>
> I find the authors' response to the "(1) Clarification of Empirical Performance" question very confusing. In the main paper, "generalizability" indicated reconstruction performance on unseen sequences. However, in the above response, the arguments are primarily focused on the **efficiency** aspect of the proposed method—claiming that its "efficient, scalable feedforward inference" enables "the scalability of directly running on unseen sequences without any per-sequence tuning."
>
> While I agree that the scalability and efficiency of this feedforward method are likely better than those of the optimization-based HOLD, if the "generalization" emphasized in the paper was actually referring to this scalability enabled by higher efficiency, I believe the overall narrative of the paper would require a significant revision.
>
> As a similar concern about this "generalization" claim was also raised by another reviewer (fKDv), I will maintain my negative rating.

---

> ### Author Response · Authors · 2025-08-07
>
> We appreciate your continued response. In our view, both the **reconstruction quality** and the **efficient scalability** for novel sequences together constitute the key aspects of generalization for high-fidelity hand-object reconstruction, rather than two unrelated points. In addition to reconstructing the hand-object pose and mesh, we also need to reconstruct flexible appearance elements such as texture, shadow, and lighting. Because existing methods such as HOLD lack **generalization capabilities** for **unseen appearance details**, they need to require a lot of post-optimization stepsand time to fit these appearance details (HOLD ~1 Hours Vs. Ours ~30 Seconds for a clip with 300 frames). Therefore, in the photorealistic hand-object reconstruction task, we argue that efficient scalability for novel sequences is also one of the important criteria for generalization ability.
>
> Accordingly, in the main paper, our narrative addresses **both** the modeling quality on unseen sequences and scalability enhancement over sequence-specific methods (e.g., in Line 27, reducing novel-scene setup time from \~1 hour to \~0.1 seconds without any post-training adaptation). To support this claim, we design comprehensive experiments from two perspectives:
> (1) Tables 2 and 4 evaluate the reconstruction and rendering performance on unseen sequences without test-time tuning; and
> (2) in Table 3, we demonstrate the comparable performance in contrast to per-sequence optimization approaches. ﻿
> ﻿
>
> In Tables 2 and 4, we achieve **leading performance on unseen sequences** compared with monocular reconstruction/rendering baselines **under the same generalization setting** (i.e., tested on novel frames without test-time tuning). In particular, in Table 4, we re-implement sequence-specific methods to investigate their performance on unseen sequences. The results highlight their limited generalization capability without per-sequence tuning. ﻿
> ﻿
>
>
> The reason we emphasized efficiency more in our response is to clarify your earlier concern regarding the seemingly comparable performance to HOLD: HOLD operates on seen sequences and requires per-sequence optimization, whereas our method offers highly efficient scalability. Additionally, we have previously clarified a similar concern brought up by Reviewer fKDv, and we hope our earlier response was helpful in that context.

---

> ### Comment · Reviewer_9ss3 · 2025-08-08
>
> I thank the authors for their response, and some of my questions have been resolved. However, I believe the overall narrative in both the main paper and the rebuttal is not very clear, and I am concerned that the paper may require a major revision to improve its clarity.
>
> In particular:
>
> 1. In the rebuttal, “generalization” is framed as a combination of reconstruction quality and efficiency, but this framing is not clearly presented in the main paper.
>
> 2. I remain unconvinced by the responses to my concern regarding the “seemingly comparable [or worse] performance to HOLD.” The authors addressed this by stating (i) that “HOLD lacks generalization capabilities for unseen appearance details,” yet the comparison in question was about shape modeling, not appearance, and (ii) by highlighting efficiency, which I agree is an advantage, but which does not address my specific question about comparable or worse reconstruction quality. I keep raising this point because I believe HOLD is the most competitive baseline among those in Tables 2 and 4, as most of the others were not originally designed for the same hand–object reconstruction task.

---

> > ### Author Response · Authors · 2025-08-08
> >
> > We appreciate your continued engagement and constructive feedback.
> > Regarding shape modeling, we believe that the **comparison under the same experimental setting** in Table 2, particularly against **monocular hand-object reconstruction methods** (e.g., MOHO-CVPR24, designed for 2D-supervised hand-held object reconstruction, closest to our setting), more convincingly demonstrates the performance advantage of our method. In particular, both MOHO and SDF-based methods are specifically designed for interaction reconstruction and testing in unseen sequences, making them more appropriate baselines for assessing our method's capability.
> >
> > In contrast, HOLD relies on additional assumptions and sequence-specific cues, making it a less comprehensive measurement compared with MOHO and SDF-based methods. For example, clip-dependent SfM (which requires a delicate balance in frame numbers, as well as smooth and continuous motion trajectories) and temporal frame embeddings.
> >
> >
> > When we attempt to neutralize these setting differences in Table 4, our method demonstrates significant improvements. These comparisons with photo-realistic rendering baselines also verify that beyond shape and pose, our approach generalizes well to **novel appearances** across unseen sequences.
> >
> >
> > We thank you again for your emphasis on the comparison with HOLD, which encourages us to conduct further analysis and discussion in the main paper. Nonetheless, we earnestly hope that our responses help alleviate your concerns and show that our current experimental results reasonably support the claimed generalization performance.

---

### Official Review · Reviewer_iQZL · 2025-07-03

**Clarity:** 2
**Significance:** 2
**Originality:** 2
**Rating:** 4
**Confidence:** 3

**Summary:**

The paper proposes a method, HOGS, for generalizable 3D hand-object reconstruction, based on 3D appearance and  pose priors learned from 2D hand-object interaction images. The method aims to disentangle base geometry from transient lighting and motion-induced appearance changes, improving generalization across diverse environments and motions. The framework includes two learnable modules, V-PM a transformer-based image encoder which disentangles appearance-invariant geometry from learnable transient variations and G-PM a 3D NN-based module used for object pose refinement. Experimental results show that HOGS outperforms state-of-the-art methods in monocular hand-object reconstruction and photo-realistic rendering.

**Questions:**

My questions are also included in the weaknesses section of this review. I would be glad to get a more detailed explanation of the training and inference process by the authors.

**Ethical Concerns:**

["NO or VERY MINOR ethics concerns only"]

**Final Justification:**

After reading the authors' rebuttal (including the provided experimental results and explanations), as well as the discussions with other reviewers, my initial concerns have been covered and thus, I raise my rating to Borderline Accept.

**Limitations:**

yes

**Paper Formatting Concerns:**

No formatting concerns

**Quality:**

2

**Strengths And Weaknesses:**

Strengths:
- The paper proposes a method from scene and motion generalizable 3D hand-object reconstruction from images based on 3DGS representation and demonstrates improved performance compared to SOTA.

- The paper includes ablations that help understand certain aspects of the framework.

- Once trained the modules do not require per-scene fine-tuning, reducing the required time for 3D reconstruction in novel environments. However, some parts of their training and inference are unclear as mentioned below.

Weaknesses:
- The design of V-PM is not clearly motivated or explained in the paper. How is transient attribute disentanglement assured by the design of Zv? Is the simple parameter combination method utilized, (addition or multiplication) of optimized parameters and their inferred offsets enough to disentangle attributes? Also, a spherical harmonics representation is still is optimized for color, how are illumination and motion related color variations split between optimized and inferred parameters?

- The training process has not been made clear to me after reading the paper. Based on the authors words, HOGS jointly optimizes deformable 3D Gaussian primitives for each hand and each object, while learning generalizable interaction priors to refine hand-object modeling. How does this process work as it requires per-scene/image optimization of 3DGS parameters and cross-scene learning of the generalizable modules?

- The inference part of the method is also not well explained as far as I can understand.
According to the authors, HOGS adapts to new scenes without the need for per-scene fine-tuning. Does this refer only to the two generalizable modules or all parts of the model? If so, how are the base 3DGS attributes inferred for the hands and objects? Is 3DGS optimization employed for inference, also utilizing V-PM and  G-PM for improved generalization and disentanglement? This is not well explained in the paper.

- G-MP updates the object poses only, while the hand poses are initialized by a 3D hand pose estimation method and left frozen, this is also visible is Figure 3 where the hand poses do not change regardless if G-PM is used or not, while the hand poses are not detailed compared to the GT. This can be problematic when the hand pose estimation produces inaccurate results.

- The authors introduce many losses in the supplementary material for which they do not offer ablation studies of a discussion of their relative effect.

- Experiments miss comparison with a recent 3D hand-object reconstruction method EasyHOI [*], which uses Large Reconstruction Models and hand-object optimization to reconstruct hand-object interactions.

[*] Liu Y, Long X, Yang Z, Liu Y, Habermann M, Theobalt C, Ma Y, Wang W. EasyHOI: Unleashing the Power of Large Models for Reconstructing Hand-Object Interactions in the Wild. In Proceedings of the Computer Vision and Pattern Recognition Conference 2025 (pp. 7037-7047).

---

> ### Author Rebuttal · Authors · 2025-07-29
>
> We thank the reviewer for recognizing the novelty, performance improvements, and the advantage of not requiring per-scene fine-tuning. Below, we provide detailed clarifications to address the concerns regarding the training and inference procedures.
>
> **Clarification of V-PM**
> The V-PM module is designed to disentangle transient, context-dependent visual effects (e.g., lighting variations, occlusions) from stable, geometry-related attributes. It achieves this via inferred offsets predicted from ViT-based visual features, which inject rich contextual cues into the parameter space. Rather than simple additive noise, these offsets enable conditional modulation of Gaussian attributes based on visual modality, promoting disentanglement.
> Take color modeling as an example, SH coefficients handle angular reflectance but fail to capture transient effects like global illumination, cast shadows, or motion-induced highlights. To address this, we introduce a visual-dependent color component, adaptively blended with SH via a learnable gate. Table 7 (ID 4) demonstrates the render quality improvement with this adaptive appearance modeling; Figure 3 shows the visual-driven modulation captures fine-grained transient effects (e.g., hand-induced shadows, dynamic reflections) that static baselines cannot reproduce.
>
> **Details of the Training and Inference Process**
> HOGS is trained end-to-end on a large collection of monocular interaction sequences that include mixed objects, motions, and environments. Training data indeed covers multiple objects, but focuses more on variation in interaction and scene context. Specifically, on DexYCB, we sample 147,526 monocular images from over 1,000 sequences covering 10 subjects and 20 objects.
> Through end-to-end training on diverse sequences with varying motions and environments, HOGS learns scene modulation capabilities. In this way, when presented with unseen sequences, HOGS can adaptively modulate the hand-object assets from monocular images in a purely feed-forward manner, without per-sequence optimization.
>
> For the inference procedure, we clarify that HOGS is intended to ***generalize to unseen scenes and motions rather than to synthesize entirely new object categories***. Accordingly, during inference, HOGS takes monocular images from unseen sequences with novel motions and viewpoints, and directly produces scene‑adapted outputs in a purely feed‑forward manner, without any fine‑tuning.
>
> **Clarification of Pose Refinement**
> In G-PM, we focus solely on refining the object pose, as it critically impacts reconstruction quality. Hand poses are not updated within G-PM; instead, hand articulation is represented using a learnable LBS weight during hand Gaussian modeling, which avoids the complexity of jointly optimizing both hand and object poses.
>
> ---
>
> **Ablation for Loss Functions**
>
> We appreciate the reviewer’s suggestion to further analyze the loss terms. In response, we conducted ablation studies on the mask loss, perceptual loss, and contact/penetration losses. The results show that removing any of these losses leads to performance degradation, particularly in terms of structural and perceptual quality metrics. We will include the complete results in the final version.
>
> | Methods                            | PSNR↑  | SSIM↑  | LPIPS↓  |
> |----------------------------------|--------|--------|---------|
> | w/o $\mathcal{L}_{mask}$          | **31.13**  | 0.9720 | 27.51   |
> | w/o $\mathcal{L}_{perc}$          | 31.12  | 0.9720 | 27.48   |
> | w/o $\mathcal{L}_{cont+pen}$  | 31.09  | 0.9727 | 26.93   |
> | Full model                  | 31.12 | **0.9728** | **26.83** |
>
> ---
>
> **Comparison with More Recent Methods**
>
> EasyHOI is a zero-shot hand-held mesh reconstruction method. It leverages the generative capability of LRMs to model arbitrary object categories. However, in this paper, we focus more on the complexity of scenes during interactions, involving irregular object orientations, frequent motion changes, and dynamic lighting effects. Such scene complexity poses significant challenges for accurate mesh generation in zero-shot models. In contrast, HOGS is explicitly trained on diverse interaction sequences, enabling hand-object assets to generalize across varied motions and environments. This fundamental difference leads to a significant performance gap, as shown below:
>
> | Methods   | CD↓    | F@5↑   | F@10↑  |
> |-----------|--------|--------|--------|
> | EasyHOI   | 1.628  | 0.134  | 0.253  |
> | Ours      | **0.085** | **0.700** | **0.909** |
>
> > *We follow the evaluation protocol of EasyHOI and test on the same 500 images from DexYCB. We report Chamfer Distance and F‑score with 5 mm and 10 mm thresholds.*
>
> Since a fair comparison is NOT applicable between these two settings, we omitted EasyHOI in the main paper. We appreciate the reviewer’s suggestion to include more recent methods. ***To this end, we additionally report a comparison with the latest 3D-supervised SOTA monocular hand-held object reconstruction approach, HORT (concurrent work, ICCV 2025 accepted)*** [1], where our method still demonstrates superior object reconstruction performance on DexYCB.
>
> | Methods | F@5↑ | F@10↑ |
> |---------|-------|-------|
> | gSDF    | 0.44  | 0.71  |
> | HORT    | 0.63  | 0.85  |
> | Ours    | **0.79**  | **0.92**  |
> > *Results of HORT are cited from its original paper. Chamfer Distance is omitted due to inconsistent computation protocols.*
> ---
>
> **References**
>
> [1] Chen Z. et al., "HORT: Monocular Hand-held Objects Reconstruction with Transformers," *ICCV*, 2025.

---

> ### Author Response · Authors · 2025-08-08
>
> We sincerely appreciate the time and effort you have invested in reviewing our paper and providing insightful feedback. As the discussion period is drawing to a close, we want to ensure that our rebuttal has comprehensively addressed your concerns. We are keen to receive any further feedback you may have and are prepared to make additional clarifications or modifications as needed.
> Additionally, we have supplemented the analysis with additional metrics to provide a more comprehensive comparison of SOTA 3D-supervised monocular hand-held object reconstruction methods. Thank you once again for your valuable insights. We look forward to your final thoughts.
>
>
> | Methods |  CD$_{mean}$↓| CD$_{median}$↓| F@5↑ | F@10↑ |
> |---------|-------|-------|-------|-------|
> | GraspingField    | 4.5 | 2.06|0.39  | 0.66  |
> | AlignSDF    | 3.9 | 1.83|0.41  | 0.68  |
> | gSDF    | 3.4 | 1.55|0.44  | 0.71  |
> | HORT    |1.1  |- |0.63  | 0.85  |
> | Ours    |**0.9** | **0.24**| **0.79**  | **0.92**  |
>
> >Besides the **median** Chamfer Distances reported in the main paper, we also include the **mean** values. The mean Chamfer Distance is more sensitive to outliers. Our method achieves superior reconstruction results across multiple metrics without relying on dense 3D supervision.

---

### Note · Authors · 2025-08-12

We sincerely thank all reviewers for their valuable feedback and recognition. Our method advances generalizable photorealistic hand-object modeling across different motions and environments by introducing scene-dependent modulation to hand-object Gaussians on large-scale interaction sequences. It requires no dense 3D annotations or time-consuming per-sequence tuning, achieving generalizable modeling on new sequences through efficient feed-forward inference, with modeling quality comparable to or even surpassing prior 3D-supervised and sequence-optimization methods.

We are pleased to have addressed most concerns raised by Reviewers bphE and fKDv, and hope our rebuttal has resolved Reviewer iQZL’s questions as well. Here, we provide a final summary response to Reviewer 9ss3:
> We believe that the comparison in Table 2 under consistent settings, especially with MOHO (CVPR-24) and SDF-based methods designed for unseen sequences, clearly shows the advantages of ours. HOLD relies on additional assumptions, making it a less comprehensive baseline. After accounting for these setting differences in Table 4, our method shows strong generalization to novel appearances. More details can be found in our earlier responses. We hope this addresses your remaining concerns and supports our generalization claims.

 During the rebuttal and discussion, we have further improved the paper in several aspects:

(i) **Detailed experimental procedures**. We have supplemented the descriptions of both training and inference pipelines to better illustrate the working mechanism.

(ii) **Inference efficiency**. We reported specific inference speed, providing a more comprehensive understanding of the method’s efficiency.

(iii) **Extended comparisons**. We have added evaluations against contemporaneous 3D-supervised hand-object reconstruction methods (e.g., HORT), offering a stronger empirical baseline.

(iv) **Clarification of cross-motion and cross-environment modeling**. We explicitly explain that our method leverages scene-dependent modulation to improve the adaptability of hand-object assets to novel contacts, poses, and illuminations.

(v) **Further Limitations discussion**. We explicitly acknowledge the current challenges in synthesizing entirely novel object categories.

We believe these clarifications and additions strengthen both the technical soundness and the empirical support of our work, and we appreciate the reviewers’ engagement that has helped us refine the paper.

---

### Decision · Program_Chairs · 2025-09-17

**Decision:**

Accept (poster)

**Comment:**

(a) This introduces two learnable components on top of deformable 3D Gaussians: 1) a Vision‑driven Perception Module that modulates Gaussian attributes with image features to capture transient appearance, and 2) a Geometry‑driven Pose Module to refine object pose. The method targets cross‑sequence generalization and feed‑forward inference (no per‑sequence tuning). Experimental results on DexYCB/HO3D indicate strong object reconstruction metrics and faster inference than optimization‑based pipelines.

(b) + Good inference efficiency by purely feed‑forward inferencing and no per‑sequence optimization.
     + Good Experimental results

(c) - The proposed method seems to rely on template‑based object pose and evaluation on same datasets as training, only can generalize to novel motions/environments, not novel categories, leading to weakens external generalization concerns.

(d) Three reviewers are ≥ borderline‑accept (5,4,4) and one remains borderline‑reject (3). The remaining issues (terminology/writing and the HOLD‑centric accuracy narrative) are addressable in camera‑ready without altering core methodology. Given demonstrated shape metrics under matched 2D‑supervised settings, added comparisons (HORT/EasyHOI), and a strong practical value proposition (feed‑forward, real‑time), the balance tips to poster acceptance.

(e) Three reviewers increased their rating after rebuttal. Authors Clarified scope of “generalization” in the rebuttal. Add more experimental evidence to prove their performance vs SOTAs suggested by reviewers. Generally, most reviewers provide positive feedback. Although 9ss3 remains negative primarily over the accuracy vs HOLD narrative and presentation clarity, AC believe that the writing/terminology issues are camera‑ready‑fixable, and the baseline coverage is improved  and G‑HOP’s exclusion is reasonably justified. For this reason, AC tend to accept this paper as poster.